# Text Has Curvature

**Karish Grover** [1] [2]   **Hanqing Zeng** [2]   **Yinglong Xia** [2]   **Christos Faloutsos** [1]   **Geoffrey J. Gordon** [1]

## Abstract

Does natural language text have an intrinsic curvature? Language is increasingly modeled in curved geometries—hyperbolic spaces for hierarchy, mixed-curvature manifolds for compositional structure—yet a basic scientific question remains unresolved: *what does curvature mean for text itself*, in a way that is native to language rather than an artifact of the embedding space we choose? We argue that text does indeed have curvature, and show how to *detect* it, *define* it, and *use* it. To this end, we propose **Texture**, a text-native, word-level discrete curvature signal, and make three contributions. **(a) Existence:** We provide empirical and theoretical certificates that semantic inference in natural corpora is non-flat. **(b) Definition:** We define Texture as a signed two-axis curvature of the word-in-context belief field—the *differential of reconciliation* between prefix and suffix—measuring, via a debiased Schrödinger transport divergence, whether adding context from one side *contracts* the semantic effect of context from the other side (*focus*, positive) or *expands* it into competing continuations (*fan-out*, negative). **(c) Utility:** Texture is actionable: it serves as a general-purpose measurement and control primitive enabling *geometry without geometric training*; we instantiate it on two representative tasks, improving long-context inference through curvature-guided compression and retrieval-augmented generation through curvature-guided routing. Together, our results establish a text-native curvature paradigm, making Texture practically useful.

## 1. Introduction

Language is not merely a bag of words: it expresses hierarchies, multi-way relations, and nonlocal constraints (agree-ment, coreference, long-range dependencies). A growing body of work has therefore advocated *non-Euclidean* geometric foundations for language representations, including hyperbolic and mixed-curvature modeling (Nickel & Kiela, 2017; Gu et al., 2018). Recent proposals even train large language models directly in hyperbolic spaces or with curvature-mixture architectures, motivated by the claim that text has inherent curved structure (He et al., 2025; Chen et al., 2024). However, these efforts largely treat curvature as an *architectural or embedding choice* rather than a *text-native quantity*. This leaves a foundational gap: *before we train curvature-aware models, how do we know what the curvature of natural language text is?*

**The gap: no text-native curvature formalism.** For graphs, intrinsic notions of discrete curvature are well developed. Ollivier–Ricci curvature defines curvature locally at each edge by comparing optimal-transport distances to combinatorial distances (Ollivier, 2007); it has been applied to detect communities, identify bottlenecks, and improve message-passing in GNNs (Sia et al., 2019). **No analogous formalism exists for text.** Current geometric approaches to language embed tokens or sequences onto a curved manifold and validate the choice by downstream performance (Ganea et al., 2018; Gu et al., 2018; Grover et al., 2025). The assumption that natural language text "has curvature" is never formalized, tested, or defined—curvature remains an architectural prior rather than a measurable property.

**Our approach: curvature as a property of two-sided inference.** We propose that if curvature is intrinsic to language, it should be detectable *before* committing to a curved architecture. The natural locus is the *word-in-context slot*: a position $i$ where both prefix $x_{<i}$ and suffix $x_{>i}$ constrain what can appear. A prefix suggests plausible continuations; a suffix retroactively constrains what could have occurred. The scientific object is *two-sided inference*—how these two sources of evidence interact when they inform the same latent slot meaning. Rather than embedding tokens in a chosen manifold, we represent each context side as a *belief distribution* over a finite set of plausible slot states, extracted from a frozen language model. Curvature then describes how these two evidence axes *interact*: does adding context from one side *contract* the semantic effect of context from the other (reconciliation into a single basin—*focus*),

[1]Carnegie Mellon University [2]Meta. Correspondence to: Karish Grover <karishg@cs.cmu.edu>.

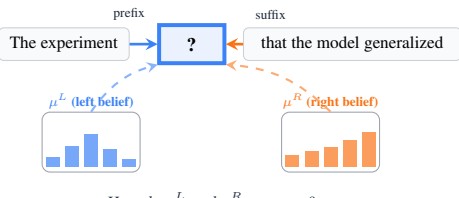

*(a)* **The word-in-context slot (local object).** A prefix and a suffix each induce a belief distribution over plausible slot fillers ($\mu^L, \mu^R$). Text curvature, in our view, is about *how two-sided evidence composes* at this local slot.

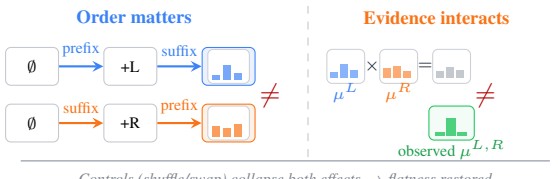

*(b)* **Flatness fails in two-sided inference.** *Left:* final belief depends on context order. *Right:* two-sided belief $\neq$ multiplicative (PoE) combination. These failures motivate curvature and underpin our existence tests (Section 4); matched controls collapse both effects.

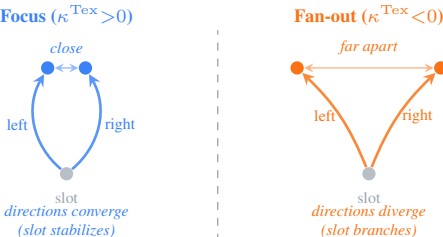

*(c)* **Texture: convergence vs. divergence.** From a slot, extending the *left* vs. *right* context leads to readings that stay *close* (*focus*, $\kappa^{\text{Tex}}>0$) or spread *apart* (fan-out, $\kappa^{\text{Tex}}<0$)—a two-axis Ricci-curvature analogue (Section 5, Fig. 3d).

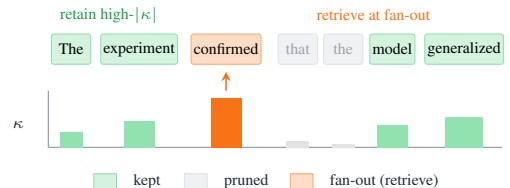

*(d)* **Curvature as control.** CURVPRUNE retains high-curvature spans under a token budget; CURVFLAG triggers retrieval at fan-out pivots. Both enable geometry-aware context management without training a curved generator (Section 6).

*Figure 1.* **Texture overview.** **(a)** Our primitive object is a two-sided slot with prefix/suffix beliefs. **(b)** Natural text exhibits non-flat two-sided belief composition (order-sensitivity and evidence interaction), motivating curvature and enabling definition-independent existence tests (Section 4). **(c)** Texture defines a text-native signed curvature from how the two-sided belief field contracts (*focus*) or expands (*fan-out*) under a neutral semantic transport geometry (Section 5). **(d)** The resulting curvature field is a general-purpose control signal: we instantiate it on pruning and retrieval, but it applies broadly to language tasks requiring context-aware resource allocation (Section 6).

or *expand* it (sustained competing alternatives—*fan-out*)? This two-direction framing—curvature as the *change* in two-sided reconciliation, not a single chord between two beliefs—yields a measurable, signed curvature field over positions in a sequence. We develop this idea in three stages:

**(a) Existence** (Section 4). Before defining curvature, we ask: *is two-sided inference over natural text effectively flat?* Flatness makes falsifiable predictions about how left and right evidence should compose as context expands. We instantiate two classical primitives—holonomy, a path-order independence condition from differential geometry (Ni, 2024), and product-of-experts combination (Hinton, 1999)—as *flatness nulls* (null hypotheses that hold if text were geometrically flat) for two-sided text inference, and test them on natural corpora against matched coherence-destroying controls. Natural text systematically violates both nulls (bootstrap-significant separation on WikiText-2 and OpenWebText); coherence-destroying controls substantially reduce both certificates toward the flatness null. This establishes that non-flat structure is a robust property of the two-sided posterior field, not an artifact of any specific curvature operator.

**(b) Definition** (Section 5). Having established non-flat

behavior, we introduce *Texture*: a text-native curvature primitive that assigns each slot a signed scalar $\kappa_i$. Using a neutral semantic transport geometry (a debiased Schrödinger/Sinkhorn divergence on slot beliefs), we measure how the two-sided belief field $\mu_i(L, R)$ deforms over a small context rectangle: $\kappa_i > 0$ (*focus*) when extending one side contracts the transport effect of extending the other, and $\kappa_i < 0$ (*fan-out*) when it expands that effect (Léonard, 2013; Cuturi, 2013; Feydy et al., 2019). This is a genuine two-axis (Ricci-type) readout, bounded in $[-1, 1]$ and well-posed on finite supports, making it practical.

**(c) Utility** (Section 6). Finally, we show that the curvature field is actionable: because Texture is computed through a frozen belief oracle, it serves as a generator-agnostic control signal for inference-time resource allocation. We introduce two plug-in utilities: CURVPRUNE, which allocates a fixed prompt budget toward high-curvature spans, and CURVFLAG, which triggers retrieval at fan-out pivots where local context is insufficient (Jiang et al., 2023b; Lewis et al., 2020). Both utilities require no geometric training of the downstream generator: curvature is estimated by a small frozen masked language model, while the generator that con-

sumes the controlled context can be any black-box LLM (we verify portability across Llama-3.1, Mistral-7B, and Qwen-2.5 in Appendix D.5). We position Texture as a complement to training curved LLMs: we exploit curved structure by *measuring* it through a frozen oracle rather than *learning* it inside the generator's representation space.

**Contributions.**

1. To our knowledge, the first text-native (extractor-only, no curved manifold) curvature primitive that is signed, computable on any frozen masked language model, and usable as a generator-agnostic control signal.

2. **Existence:** $\kappa$-independent certificates—holonomy and product-of-experts additivity (CEI), with matched controls—that two-sided inference over natural text is non-flat (Tables 4–5).

3. **Definition:** *Texture*, a signed two-axis curvature $\kappa_i^{\text{Tex}} \in [-1, 1]$ of the belief field $\mu_i(L, R)$ under a neutral transport geometry: *focus* ($> 0$) where adding one side contracts the other's effect, *fan-out* ($< 0$) where it expands.

4. **Utility:** training-free curvature-guided compression and retrieval routing that improve long-context inference under fixed token budgets.

**Conflict of Interest Disclosure.** K.G. is affiliated with both Carnegie Mellon University and Meta. H.Z. and Y.X. are employed by Meta, which develops Llama-3.1-8B-Instruct, one of the downstream generators evaluated in this paper. The Texture curvature operator itself is computed by a separate frozen belief-oracle pipeline (DistilRoBERTa and the additional masked language models studied in Appendix D.6) that is unaffiliated with the generator under control; all comparisons are reported using fixed public model checkpoints and the reproducible scripts released alongside the paper.

## 2. Related Work

**Curved representations for language and foundation models.** A growing line of work argues that language exhibits hierarchical and scale-free structure that can be modeled more naturally in non-Euclidean spaces. This view is synthesized in the recent position paper of He et al. (2025). On the modeling side, hyperbolic geometry has been explored from embedding-level approaches to end-to-end hyperbolic networks (Nickel & Kiela, 2017; Ganea et al., 2018), and Chen et al. (2024) train a hyperbolic pre-trained language model with broad gains over Euclidean baselines. These efforts validate curvature as an *architectural prior*; our work targets a different primitive: a *text-native* curvature defined and measured locally from two-sided inference without committing the generator to a curved manifold.

**Discrete curvature on graphs.** Outside NLP, a substantial literature has developed discrete curvature notions for graphs and networks. Ollivier-Ricci curvature (ORC) defines curvature via optimal-transport contraction of neighborhood measures under a random walk (Ollivier, 2007), and Forman introduced a combinatorial Ricci curvature for cell complexes (Forman, 2003), later adapted to weighted networks as an efficient edge-based curvature statistic (Sreejith et al., 2016). These notions quantify curvature of an *explicit discrete space* (graph/Markov structure) chosen by the practitioner. In language one can build auxiliary graphs (co-occurrence, syntactic, knowledge graphs), but this does not define curvature of *text itself* at the word-in-context level. Our contribution fills this gap by defining curvature directly on the *two-sided belief geometry* at a slot, rather than on a user-chosen graph.

**Entropic transport and Schrödinger bridges as a reconciliation primitive.** Entropic optimal transport provides stable, scalable computation via Sinkhorn-type scaling (Cuturi, 2013), and Schrödinger bridges define entropic interpolations by KL projection onto reference Markov paths (Léonard, 2013). These tools have also been used to study entropic curvature on discrete spaces via behavior of entropy along Schrödinger bridges (Samson, 2022). We repurpose this machinery for language: the debiased Schrödinger/Sinkhorn transport between prefix and suffix beliefs becomes a neutral semantic geometry, and Texture reads curvature from whether adding context *focuses* meaning or *fans out*.

**Long-context efficiency: compression, retrieval, and adaptive routing.** Long-context deployment motivates methods that allocate limited compute and context budget selectively. Prompt compression methods reduce cost under a token budget (Jiang et al., 2023b), while retrieval-augmented generation injects external evidence for knowledge-intensive tasks (Lewis et al., 2020). These pipelines propose heuristics or learned controllers, but do not provide a text-native geometric diagnostic of when two-sided evidence is redundant versus underdetermined. Our utilities use curvature as a signal for pruning and retrieval, leaving the generator unchanged.

## 3. Preliminaries

We collect notation and standard probabilistic/transport primitives used throughout. All paper-specific operators (e.g., *Texture* curvature) are defined later; here we only define the shared objects they are built from.

**Text, slots, and context radii.** We write a token sequence as $x_{1:n} = (x_1, \ldots, x_n)$. For index $i$, denote prefix $x_{<i} := x_{1:i-1}$ and suffix $x_{>i} := x_{i+1:n}$. For radii

$L, R \in \mathbb{N}$, we use truncated contexts $x^L_{<i} := x_{i-L:i-1}$ and $x^R_{>i} := x_{i+1:i+R}$ (clipped at boundaries), and the contextual slot $(x^L_{<i}, \square, x^R_{>i})$ where $\square$ marks a missing filler.

**Finite slot state spaces, tail bucket, and belief extractors.** All core objects live on a *finite* slot-local state space $\mathcal{S}_i$ and its simplex $\Delta(\mathcal{S}_i) := \{\mu \in \mathbb{R}^{\mathcal{S}_i}_{\geq 0} : \sum_{s \in \mathcal{S}_i} \mu(s) = 1\}$. We use one-sided boundary beliefs $\mu^L_i, \mu^R_i \in \Delta(\mathcal{S}_i)$ (prefix-only vs. suffix-only) and a two-sided belief extractor $\mathcal{B}_\leftrightarrow$ (e.g., a masked/infilling LM (Devlin et al., 2019)). On a context-radius grid, the two-sided posterior field is written $\mu_i(L, R) := \mathcal{B}_\leftrightarrow(\cdot \mid x^L_{<i}, \square, x^R_{>i}; \mathcal{S}_i) \in \Delta(\mathcal{S}_i)$. Throughout Section 5 we instantiate the one-sided boundary beliefs as $\mu^L_i := \mu_i(L_{\max}, 0)$ and $\mu^R_i := \mu_i(0, R_{\max})$, i.e. the maximal-radius slices of this two-sided field; with this convention Section 4's $\mu^{\leftarrow}_i(L) = \mu_i(L, 0)$ and $\mu^{\rightarrow}_i(R) = \mu_i(0, R)$ are the same family of objects. Because practical extractors return truncated supports (e.g., top-$k$ candidates), we use a *canonical per-slot support* with an explicit tail bucket to avoid support-change artifacts: for any belief $\mu$ on a larger ambient domain, let $\mathrm{TopK}(\mu)$ be its $k$ highest-probability states (ties broken deterministically), define $\mathcal{C}_i := \mathrm{TopK}(\mu^L_i) \cup \mathrm{TopK}(\mu^R_i)$ and $\mathcal{S}_i := \mathcal{C}_i \cup \{\mathrm{tail}\}$, and push residual mass to $\mathrm{tail}$ via $\mu(\mathrm{tail}) := 1 - \sum_{s \in \mathcal{C}_i} \mu(s)$, leaving $\mu(s)$ unchanged for $s \in \mathcal{C}_i$. We apply this map to each boundary belief and each $\mu_i(L, R)$ so that all objects share a fixed domain.

**KL divergence.** For $\mu, \nu \in \Delta(\mathcal{S})$ with $\mu \ll \nu$, the Kullback–Leibler divergence is $\mathrm{KL}(\mu \| \nu) := \sum_{s \in \mathcal{S}} \mu(s) \log \frac{\mu(s)}{\nu(s)}$. KL is used both as a discrepancy between beliefs (e.g., CEI in Section 4) and as the objective defining Schrödinger bridges (Section 5).

**Markov kernels, stationarity, and reversibility.** A Markov kernel on a finite $\mathcal{S}$ is a row-stochastic matrix $K$; it pushes forward beliefs by $(\mu K)(s') = \sum_s \mu(s) K(s, s')$. A distribution $\pi$ is stationary if $\pi = \pi K$. We call $K$ reversible w.r.t. $\pi$ if it satisfies detailed balance: $\pi(s) K(s, s') = \pi(s') K(s', s)$ for all $s, s'$. Reversibility ensures the neutral reference dynamics in Texture are left–right invariant; Appendix C.1 shows the symmetric ground cost $c_i$ induces such a reversible kernel.

**Entropic optimal transport and Sinkhorn scaling on finite supports.** For beliefs $\rho, \sigma \in \Delta(\mathcal{S})$ and a symmetric ground cost $c$, the entropic OT cost $C_\varepsilon(\rho, \sigma)$ is the regularized objective in (7); on finite supports its optimal coupling exists, is unique, and admits a multiplicative scaling form solved efficiently by log-domain Sinkhorn / matrix-scaling iterations (Cuturi, 2013). Section 5 uses its debiased (self-bias-removed) form—the symmetric Sinkhorn divergence (Feydy et al., 2019)—as a neutral semantic dissimilarity between beliefs. As optional foundations, the same Gibbs

kernel induces a reversible reference chain (stationary $\pi$, kernel $K$) and a two-step Schrödinger bridge; we develop that construction, used only for the appendix diagnostics, in Appendix C.2.

# 4. Existence: Curvature Fingerprints in Two-Sided Inference Geometry

Can we certify that natural text exhibits *non-flat* two-sided inference geometry *without* committing to any particular curvature definition? Part I answers this by proposing two *definition-independent*, falsifiable flatness nulls on the two-sided posterior field. Violations of either null are curvature certificates: they witness genuine left–right interaction in contextual belief composition. We present the nulls and their informal equivalences in the main text, with full statements and proofs in Appendix B, and we empirically falsify both nulls with coherence-destroying controls in §4.3.

## 4.1. Two-Sided Posterior Field

Fix a token sequence $x_{1:n}$ and an index $i$. For radii $L, R \in \mathbb{N}$, let $x^L_{<i} := x_{i-L:i-1}$ and $x^R_{>i} := x_{i+1:i+R}$ denote truncated left and right contexts (clipped at boundaries), and consider the slot $(x^L_{<i}, \square, x^R_{>i})$. Let $\mathcal{S}_i$ be a finite slot-local state space (e.g., top-$k$ candidates plus a tail bucket as in §3). Given a two-sided belief extractor $\mathcal{B}_\leftrightarrow$ (e.g., an infilling model), define the posterior field

$$\mu_i(L, R) := \mathcal{B}_\leftrightarrow\big(\cdot \mid x^L_{<i}, \square, x^R_{>i}; \mathcal{S}_i\big) \in \Delta(\mathcal{S}_i). \quad (1)$$

The theory in Appendix B assumes full support, which in practice is ensured by the tail bucket and/or a small $\varepsilon$-smoothing.

## 4.2. Falsifiable Flatness Nulls and Curvature Certificates

Curvature is a second-order phenomenon: in a flat geometry, incremental evidence updates should compose in a path-independent way. We formalize this intuition as two complementary, falsifiable flatness nulls on $(L, R) \mapsto \mu_i(L, R)$.

**Certificate I: Holonomy (order-sensitive evidence updates).** Fix a reference state $s_{\mathrm{ref}} \in \mathcal{S}_i$ and define log-odds coordinates

$$u_{i,s}(L, R) := \log \frac{\mu_i(L, R)(s)}{\mu_i(L, R)(s_{\mathrm{ref}})}, \qquad s \in \mathcal{S}_i. \quad (2)$$

Define the unit-square holonomy

$$\begin{aligned} \Omega_{i,s}(L, R) := \; & u_{i,s}(L+1, R+1) - u_{i,s}(L+1, R) \\ & - u_{i,s}(L, R+1) + u_{i,s}(L, R), \end{aligned} \quad (3)$$

which measures whether the one-step gain from extending $R$ depends on whether $L$ has already been extended (and vice

versa). In a flat (path-independent) regime, such incremental updates commute and $\Omega_{i,s}$ vanishes.

**Theorem 4.1** (Holonomy flatness null; see Theorem B.1). *Assume $\mu_i(L,R) \in \Delta^\circ(\mathcal{S}_i)$ on a rectangular grid domain. Then $\Omega_{i,s}(L,R) \equiv 0$ for all $s$ and all unit squares if and only if there exist functions $\alpha_{i,s}(\cdot)$ and $\beta_{i,s}(\cdot)$ such that $u_{i,s}(L,R) = u_{i,s}(0,0) + \alpha_{i,s}(L) + \beta_{i,s}(R)$. Equivalently, left and right evidence contribute additively in log-odds.*

**Curvature certificate (Holonomy).** If $\Omega_{i,s}(L,R) \neq 0$ for some $s$ and adjacent cell $(L,R)$, then the additive-separability flatness null fails. We summarize holonomy per slot by an RMS magnitude over a small grid $\mathcal{G}$ (and optionally probability weights; see Appendix B.1):

$$h_i := \left( \frac{1}{|\mathcal{G}|} \sum_{(L,R)\in\mathcal{G}} \sum_{s\in\mathcal{S}_i} w_{i,s}(L,R)\,\Omega_{i,s}(L,R)^2 \right)^{1/2},$$
(4)

where $w_{i,s}(L,R) \propto \mu_i(L+1,R+1)(s)$ (normalized over $s$) emphasizes probable states (Appendix B.1).

**Certificate II: Evidence additivity (Product-of-Experts) and CEI.** A second flatness prediction is that two-sided evidence should combine additively in log space. For fixed radii $(L,R)$, define one-sided boundary beliefs $\mu_i^{\leftarrow}(L) := \mu_i(L,0)$, $\mu_i^{\rightarrow}(R) := \mu_i(0,R)$, and base belief $\mu_i^{(0)} := \mu_i(0,0)$. The Product-of-Experts reconstruction is

$$\mu_i^{\text{PoE}}(L,R)(s) \propto \frac{\mu_i^{\leftarrow}(L)(s)\,\mu_i^{\rightarrow}(R)(s)}{\mu_i^{(0)}(s)}.$$
(5)

We measure deviation from this null by the Contextual Evidence Interaction statistic

$$\text{CEI}_i(L,R) := \text{KL}\big(\mu_i(L,R) \,\big\|\, \mu_i^{\text{PoE}}(L,R)\big) \geq 0.$$
(6)

**Theorem 4.2** (PoE flatness null; see Theorem B.2). *Assume $\mu_i(L,R) \in \Delta^\circ(\mathcal{S}_i)$. Then $\text{CEI}_i(L,R) = 0$ if and only if $\mu_i(L,R) = \mu_i^{\text{PoE}}(L,R)$. In particular, under an additive-evidence/conditional-independence null where left and right context contribute independent log Bayes factors given the slot value, PoE holds and $\text{CEI}_i(L,R) = 0$.*

**Curvature certificate (PoE/CEI).** If $\text{CEI}_i(L,R) > 0$, then the additive-evidence flatness null fails, witnessing non-additive left–right interaction in two-sided inference. Holonomy probes *order-sensitivity* around loops in $(L,R)$, while CEI probes *non-additivity* of evidence composition. Both certificates are definition-independent: they remain meaningful regardless of how one later defines Texture.

### 4.3. Empirical Falsification on Natural Corpora

We test whether natural text violates these flatness nulls using a frozen infilling model $\mathcal{B}_{\leftrightarrow}$ instantiated by

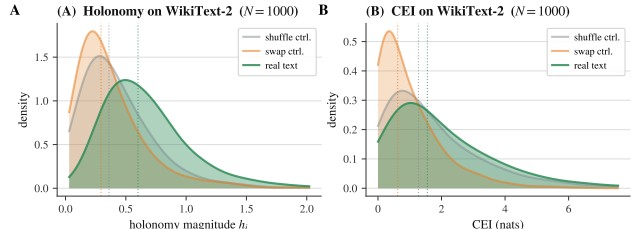

*Figure 2.* **Empirical falsification of flatness nulls (Panels A–B).** Distributions are over 1000 slots per corpus (WikiText-2 and OpenWebText) using `distilroberta-base`. **(A)** Holonomy magnitude $h_i$ is consistently higher on natural text than on suffix-swap and local-shuffle controls, indicating order-sensitive evidence updates in two-sided inference. **(B)** CEI shows analogous separation, indicating non-additive evidence composition relative to the PoE null. Full diagnostics, medians/effect sizes, and bootstrap CIs appear in Appendix B.4.

`distilroberta-base` (a distilled RoBERTa model; Sanh et al., 2019; Liu et al., 2019). We evaluate 1000 randomly sampled slots per corpus on the radius grid $L, R \in \{0, 1, 2, 4, 8\}$ using a fixed per-slot support $\mathcal{S}_i$ (with a tail bucket; details in Appendix B.3). We compare *natural text* to two coherence-destroying controls: (i) **suffix-swap**, which exchanges the right contexts of matched slots, and (ii) **local-shuffle**, which permutes tokens within the right window while preserving its multiset. Both preserve surface statistics while disrupting two-sided coherence.

Figure 2 summarizes both certificates on WikiText-2 (Merity et al., 2016) and OpenWebText (Gokaslan & Cohen, 2019). Natural text exhibits substantially larger holonomy magnitude $h_i$ and larger CEI than either control, refuting both commuting-update and additive-evidence nulls. Additional diagnostics—a joint holonomy/CEI density, a slot-paired real-vs-swap holonomy scatter, and full numeric summaries with bootstrap CIs—appear in Appendix B.4. Together, these two falsifiable certificates establish an existence fact independent of our later definitions: two-sided inference geometry over natural text is generically non-flat. Section 5 (Definition) defines Texture to measure this interaction; Section 6 (Utility) applies the resulting curvature as a control signal.

## 5. Texture: Contextual Ricci Curvature

Section 4 established—via definition-independent, falsifiable certificates—that two-sided inference at a word-in-context slot exhibits intrinsic second-order interaction. We now define *Texture*, a text-native curvature primitive. We define curvature at the smallest unit with two independent evidence axes—the slot $(x_{<i}, \square, x_{>i})$, which admits a left belief (prefix) and a right belief (suffix)—precisely where "flat" additive composition can fail. Texture assigns each slot $i$ a signed scalar $\kappa_i \in [-1, 1]$ with an interpretable

sign: *focus* ($\kappa_i > 0$), where adding context from one side *contracts* the semantic effect of adding context from the other side, concentrating meaning into a basin, and *fan-out* ($\kappa_i < 0$), where the two sides *expand* each other's marginal effect, inducing branching among alternatives.

The definition is constrained by two design principles. First, curvature is the *differential of reconciliation*. Prefix and suffix give two contextual opinions about the slot; a single bridge between $\mu_i^L$ and $\mu_i^R$ captures only their reconciliation *tension*, whereas a signed curvature—like Ricci curvature— is detected by how motion along one axis changes motion along the other, not from a single chord. We therefore read Texture from the two-sided posterior *field* $F_i : (L, R) \mapsto \mu_i(L, R)$ over a small context rectangle—the same object whose non-flatness Part I certifies—measuring how the reconciliation *changes* as each side of context is extended, rather than scoring a single boundary pair $(\mu_i^L, \mu_i^R)$. Second, the geometry of that field must be measured by a *neutral* notion of semantic motion. We measure semantic motion with a finite-state entropic-transport *divergence*: a debiased Sinkhorn divergence under the slot cost $c_i$, built from the same reversible Schrödinger-bridge kernel (Léonard, 2013; Feydy et al., 2019; Appendix C.2). Texture is then the signed *transport contraction* of the field across context rectangles: the Schrödinger bridge supplies the neutral geometry of reconciliation, while the *sign* is read from how that reconciliation contracts or expands.

### 5.1. Slot State Space and Boundary Beliefs

Fix a token sequence $x_{1:n}$ and a slot index $i$. We model the contextual blank $(x_{<i}, \square, x_{>i})$ as two observations of an unobserved slot state $Z_i \in \mathcal{S}_i$, where $\mathcal{S}_i$ is a *finite state space* local to slot $i$. Finiteness is a computational design choice: Texture is meant to be computed from truncated (top-$k$) supports produced by frozen models.

A *left boundary belief* is $\mu_i^L \in \Delta(\mathcal{S}_i)$ intended to approximate $P(Z_i = \cdot \mid x_{<i})$, and a *right boundary belief* is $\mu_i^R \in \Delta(\mathcal{S}_i)$ intended to approximate $P(Z_i = \cdot \mid x_{>i})$. Per §3, the primary object is the two-sided posterior *field* $\mu_i(L, R)$ evaluated on a radius grid $\mathcal{G} = \{0, 1, 2, 4, 8, 16\}^2$ (the same field Part I certifies as non-flat); the extreme anchors $\mu_i^L := \mu_i(L_{\max}, 0)$ and $\mu_i^R := \mu_i(0, R_{\max})$ fix the canonical support below. Texture is *extractor-dependent* in magnitude but *generator-agnostic* in deployment: the curvature signal feeds any downstream LLM without changing its representation geometry. Texture can be instantiated longitudinally (token-filler state spaces) or transversally (relational state spaces over AMR/OpenIE-style graphs); we present the relational instantiation and its validation in Appendix C.6 and the transversal experiments in Appendix D.8.

**Canonical support with a tail bucket.** To avoid support-mismatch artifacts from truncation, we define a canonical per-slot support by augmenting the union of top-$k$ candidates with a tail bucket state. Let $k \in \mathbb{N}$ and let $\mathrm{TopK}(\mu)$ return the $k$ highest-probability states under $\mu$ (ties broken deterministically). Define $\mathcal{C}_i := \mathrm{TopK}(\mu_i^L) \cup \mathrm{TopK}(\mu_i^R)$ and $\mathcal{S}_i := \mathcal{C}_i \cup \{\text{tail}\}$. We push forward beliefs to $\mathcal{S}_i$ by assigning leftover mass to tail: $\mu_i^L(\text{tail}) := 1 - \sum_{s \in \mathcal{C}_i} \mu_i^L(s)$ and $\mu_i^R(\text{tail}) := 1 - \sum_{s \in \mathcal{C}_i} \mu_i^R(s)$, while keeping $\mu_i^L(s), \mu_i^R(s)$ unchanged for $s \in \mathcal{C}_i$. This avoids renormalization and ensures downstream objects are well-posed on a fixed domain.

### 5.2. Neutral Semantic Transport Geometry

Texture requires a neutral reference notion of semantic motion on $\mathcal{S}_i$, specified by a symmetric ground cost $c_i$ with $c_i(s, s) = 0$; in longitudinal mode we induce it from frozen embeddings, $c_i(s, s') := \|\hat{e}(s) - \hat{e}(s')\|_2^2$ with $\hat{e}(s) := e(s)/\|e(s)\|_2$ (transversal $c_i$ is defined on relational nodes, Appendix C.6). For a temperature $\varepsilon > 0$, the entropic optimal-transport cost between beliefs $\rho, \sigma \in \Delta(\mathcal{S}_i)$ under $c_i$ is

$$C_\varepsilon(\rho, \sigma) := \min_{\gamma \in \Pi(\rho, \sigma)} \langle \gamma, c_i \rangle + \varepsilon \, \mathrm{KL}(\gamma \| \rho \otimes \sigma), \quad (7)$$

the standard regularized OT objective, solved by log-domain Sinkhorn iterations on the Gibbs kernel $e^{-c_i/\varepsilon}$ (Cuturi, 2013; the reversible kernel $K_i$ and its stationary $\pi_i$ used elsewhere are detailed in Appendix C.1). The *debiased Sinkhorn divergence*

$$\mathsf{d}_i(\rho, \sigma) := \sqrt{\left[ C_\varepsilon(\rho, \sigma) - \tfrac{1}{2} C_\varepsilon(\rho, \rho) - \tfrac{1}{2} C_\varepsilon(\sigma, \sigma) \right]_+}$$

$$(8)$$

removes the entropic self-bias and gives a nonnegative, symmetric, semantically aware *divergence* with $\mathsf{d}_i(\rho, \rho) = 0$ (Feydy et al., 2019)—the transport dissimilarity Texture uses to measure the geometry of the posterior field. (We say "divergence" rather than "metric": we do not assume a triangle inequality.)

### 5.3. From Field Geometry to Curvature: Texture

Texture reads signed curvature from how the two-sided posterior field $F_i : (L, R) \mapsto \mu_i(L, R)$ contracts or expands across a small context rectangle, measured by the transport divergence $\mathsf{d}_i$. On an adjacent cell of the radius grid with corners $\mu_{00} = \mu_i(L, R)$, $\mu_{10} = \mu_i(L+\delta_L, R)$, $\mu_{01} = \mu_i(L, R+\delta_R)$, $\mu_{11} = \mu_i(L+\delta_L, R+\delta_R)$, the *left* increment before/after extending the right context is $\ell_L^0 = \mathsf{d}_i(\mu_{00}, \mu_{10})$, $\ell_L^R = \mathsf{d}_i(\mu_{01}, \mu_{11})$, and symmetrically $\ell_R^0 = \mathsf{d}_i(\mu_{00}, \mu_{01})$, $\ell_R^L = \mathsf{d}_i(\mu_{10}, \mu_{11})$. The signed local contraction score is

$$s_i(L, R) := \tfrac{1}{2} \left[ \frac{\ell_L^0 - \ell_L^R}{\ell_L^0 + \ell_L^R + \eta} + \frac{\ell_R^0 - \ell_R^L}{\ell_R^0 + \ell_R^L + \eta} \right] \in [-1, 1].$$

$$(9)$$

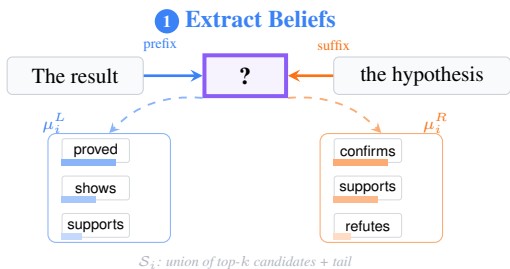

*(a)* **Boundary beliefs (slot input).** For a slot $(x_{<i}, \square, x_{>i})$, a frozen belief extractor yields prefix and suffix distributions $\mu_i^L$ and $\mu_i^R$ over a finite candidate set $\mathcal{S}_i$ (Top-$k$ union + `tail`).

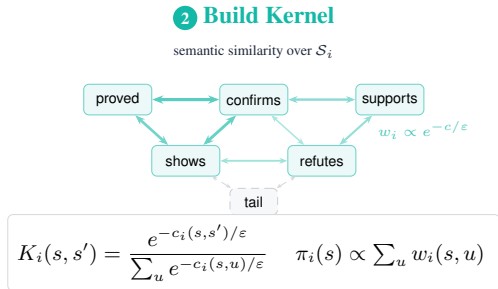

*(b)* **Neutral semantic motion kernel.** A symmetric slot-local semantic cost $c_i$ defines affinities among candidates in $\mathcal{S}_i$; row-normalization yields a Markov kernel $K_i$ (with stationary $\pi_i$) that supplies the neutral notion of "semantic drift" underlying the transport divergence.

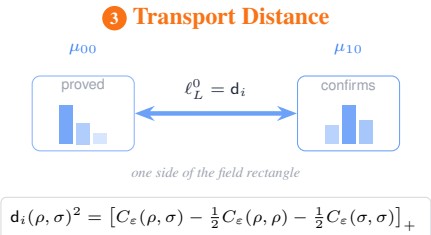

*(c)* **Neutral semantic transport.** The cost $c_i$/kernel $K_i$ induce an entropic transport cost $C_\varepsilon$; its debiased Sinkhorn divergence $\mathsf{d}_i$ is a symmetric, semantically aware dissimilarity between beliefs (with $\mathsf{d}_i(\rho, \rho) = 0$). Each $\mathsf{d}_i$ measures *one* side of the posterior-field rectangle (here the left-increment side $\ell_L^0 = \mathsf{d}_i(\mu_{00}, \mu_{10})$), not a single endpoint chord.

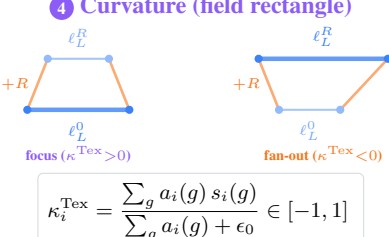

*(d)* **Signed curvature from field contraction.** Each cell is a $2{\times}2$ patch of $\mu_i(L, R)$: the horizontal sides are the left-update $\mathsf{d}_i$ ($\ell_L^0$ before, $\ell_L^R$ after adding right context, $+R$). When $+R$ *shrinks* the left-update ($\ell_L^R < \ell_L^0$) the cell is *focus* ($\kappa^{\text{Tex}} > 0$); when it *grows* it, *fan-out* ($\kappa^{\text{Tex}} < 0$). $\kappa_i^{\text{Tex}}$ is the activity-weighted signed contraction over cells.

*Figure 3.* **Texture operator pipeline.** Given a word-in-context slot, we (a) extract boundary beliefs $\mu_i^L$, $\mu_i^R$ on $\mathcal{S}_i$, (b) build a neutral semantic cost / kernel from embeddings, (c) turn it into a debiased Sinkhorn transport distance $\mathsf{d}_i$ between beliefs, and (d) read off signed curvature from how the two-sided posterior field $\mu_i(L, R)$ contracts or expands across context rectangles (focus vs fan-out).

Intuitively, $s_i > 0$ when adding the opposite context *shrinks* the marginal effect of more evidence (the two sides stabilize the slot, *focus*); $s_i < 0$ when it *enlarges* that effect (the two sides branch, *fan-out*); and $s_i \approx 0$ on a locally flat rectangle. Each term is a *relative* length change, so the bound $s_i \in [-1, 1]$—and hence $\kappa_i \in [-1, 1]$—is intrinsic, not an imposed clip. Texture aggregates $s_i$ over the adjacent cells $g \in \mathcal{G}$ of the grid, weighting each cell by an activity gate $a_i(g) = \tanh\big((\|\Omega_i(g)\| + \lambda\,\mathrm{CEI}_i(g))/\tau_a\big)$ built from the *same* Part I non-flatness certificates (holonomy $\Omega_i$ and CEI):

$$\kappa_i^{\text{Tex}} := \frac{\sum_{g \in \mathcal{G}} a_i(g)\, s_i(g)}{\sum_{g \in \mathcal{G}} a_i(g) + \epsilon_0} \in [-1, 1]. \qquad (10)$$

We write $\kappa_i := \kappa_i^{\text{Tex}}$ throughout. The guard $\epsilon_0 > 0$ keeps the denominator strictly positive, so $\kappa_i$ is well-defined even with no active cells ($\sum_g a_i(g) = 0$, reading 0; likewise 0 when every cell is below numerical thresholds). The activity-weighted form is canonical because its *sign* comes from transport contraction while its *confidence* comes from

the same certificates that establish curvature exists (boundedness and the flat-rectangle null: Appendix C.3). This is a two-axis (Ricci-type) readout, not an endpoint-disagreement statistic—left/right disagreement alone does not fix the sign—and not a salience score for the realized token, but a signed second-order interaction of the belief field around it.

**Empirical sign distribution.** On natural English text both regimes occur at slot resolution; their balance is corpus-dependent rather than imposed by the definition, shifting markedly across corpora (Appendix D.7, Figures 10 and 11).

### 5.4. Computation

For each slot, the field $\{\mu_i(L, R)\}$ is evaluated on the radius grid $\mathcal{G} = \{0, 1, 2, 4, 8, 16\}^2$ (one masked-LM forward per corner, batched); each cell score (9) needs the debiased Sinkhorn divergences (8) between adjacent corners—small log-domain matrix-scaling problems on $c_i$, memoized across shared corners. A tiny smoothing of beliefs and kernel ensures full support, and the resulting scores

are stable on finite supports (Theorem C.6; algorithm in Appendix C.5).

**Transversal instantiation.** The same construction extends beyond token fillers to *relational* state spaces—nodes from dependency parses and OpenIE-style triples—using identical transport machinery, a complementary relational view we develop in Appendix C.6 (rank agreement and cost in Appendix D.8).

## 6. Utility: Curvature as an Inference-Time Control Primitive

Section 4 shows two-sided inference over natural text is non-flat, and Section 5 defines $\kappa_i$ as a signed scalar summarizing that non-flatness. We use the curvature field as a *generator-agnostic* inference-time control signal: computed by a frozen masked-LM oracle, it allocates budget or triggers retrieval for an unrelated downstream generator, addressing a common question—*where is local context sufficient, and where is more evidence needed?*

**CURVPRUNE: budgeted prompt construction.** Given a query $q$, a long context $x$, and a token budget $B$, CURVPRUNE (Algorithm 2; span scoring and guard bands in Appendix D.2) computes the curvature field along $q\|x$, scores spans by aggregated magnitude, and greedily selects spans (with an optional guard band) until the budget is exhausted. The span score is the mean per-slot magnitude with optional sign weighting,

$$s(I) \;=\; \frac{1}{|I|} \sum_{i \in I} \Big( w_- \, [-\kappa_i^{\mathrm{Tex}}]_+ + w_+ \, [\kappa_i^{\mathrm{Tex}}]_+ \Big), \quad (11)$$

where $[\cdot]_+ := \max(\cdot, 0)$ and $\kappa_i^{\mathrm{Tex}} \in [-1, 1]$ is the (already bounded) Contextual Ricci Texture of Section 5. Defaults $(w_-, w_+) = (1.0, 0.25)$ favor fan-out regions, since underdetermined positions are the ones most likely to benefit from preserved context.

**CURVFLAG: curvature-guided retrieval routing.** For retrieval-augmented pipelines, CURVFLAG (Algorithm 3) maps the slot-local fan-out mass into a retrieval budget. Concretely, evaluating curvature on a sequence $z \in \{q, \, q\|\mathrm{draft}(q)\}$, define the *fan-out mass*

$$M_-(z) \;:=\; \sum_{i \in \mathcal{I}(z)} \big[ -\kappa_i^{\mathrm{Tex}}(z) \big]_+ \quad (12)$$

and set the retrieval budget by an affine clip

$$k(M_-) \;:=\; \mathrm{clip}\big( k_{\min} + \alpha \, M_-, \; k_{\min}, \; k_{\max} \big), \quad (13)$$

with the clip (13) optionally falling back to a full-context pathway when $M_-$ saturates above $k_{\max}$. Anchor spans for query augmentation are extracted from the top-$m$ fan-out slots in $z$ (Appendix D.3).

**Complexity.** Per slot, the dominant cost is the batched frozen-oracle evaluation of the context-radius grid for $\mu_i(L, R)$ (one forward per grid node). The Sinkhorn steps are small matrix-scaling problems on the top-$k$ support ($|\mathcal{S}_i| \leq 51$), memoized across corner-sharing queries, so they are negligible relative to the oracle forwards; sparse slot sampling and cache reuse amortize cost across nearby slots and budget sweeps (Appendices D.4, E.1, D.1).

## 7. Experiments

We evaluate the *utility* of Texture as an inference-time control signal for long-context workloads: whether curvature is actionable for budgeted context selection and retrieval routing (existence and definition are evaluated in their own sections). All curvature quantities are computed using a frozen masked-language oracle (distilroberta-base), while downstream generation uses Llama-3.1-8B-Instruct (Grattafiori et al., 2024) without fine-tuning (compute environment in Appendix E.1); we additionally validate generator portability across Mistral-7B (Jiang et al., 2023a) and Qwen-2.5-7B (Bai et al., 2025) in Appendix D.5. We use five representative LongBench tasks spanning multi-hop QA, long-document QA, and summarization (Bai et al., 2023): HotpotQA (Yang et al., 2018), 2WikiMultiHopQA (Ho et al., 2020), Qasper (Dasigi et al., 2021), GovReport (Huang et al., 2021), and QMSum (Zhong et al., 2021), reporting mean F1 / ROUGE-L with bootstrap intervals over per-example deltas (statistical protocol in Appendix E.3).

**Baselines.** For pruning we compare against heuristics (random, recency, head+tail), query-aware BM25 (Robertson & Zaragoza, 2009), and learned compression (Selective Context (Li et al., 2023), LLMLingua (Jiang et al., 2023b)); for routing, against fixed-$k$ retrieval, FLARE (Jiang et al., 2023c), and Self-Route (Li et al., 2024). Baseline configurations are in Appendix E.6.

### 7.1. Longitudinal Texture: Budgeted Compression + Retrieval Routing

We instantiate two curvature-controlled utilities. CURVPRUNE selects spans to satisfy a context budget $B$ using the curvature-derived span score (11), and CURVFLAG routes between a retrieval and a long-context pathway via a fan-out statistic (12). We also evaluate paired "+Texture" variants where Texture modulates a baseline's primary signal (e.g. BM25→BM25+Texture), isolating two questions: does standalone curvature select good spans, and does it *re-allocate* a relevance-selected budget productively? Tables 1–2 report pruning at $B=2048$ and routing at a fixed cost target $T$; their rightmost $\Delta\%$ **(Tex.)** column gives the paired *Texture effect*—the relative F1 gain

*Table 1.* **Pruning on LongBench** ($B{=}2048$). F1 for QA, ROUGE-L for summarization (means over examples); token budgets are matched across methods as in Appendix E.2. The rightmost $\Delta\%$ **(Tex.)** column is the *Texture effect*—mean relative F1 gain on multi-hop QA over BM25.

| Method | 2Wiki | GovRep | HpQA | Qasper | QMSum | $\Delta\%$ (Tex.) |
|---|---|---|---|---|---|---|
| *Reference* | | | | | | |
| LC | 8.1 | 21.1 | 19.1 | 8.9 | 16.1 | – |
| *Heuristic* | | | | | | |
| Random | **16.1** | **20.6** | 20.5 | 6.5 | 14.3 | – |
| Recency | 5.8 | 18.8 | 17.0 | 6.5 | 13.8 | – |
| Head+Tail | 10.9 | 20.2 | 17.9 | 6.7 | 14.0 | – |
| *Query-aware / learned* | | | | | | |
| BM25 | 9.4 | 19.8 | 28.1 | 8.4 | 15.1 | *ref* |
| SelectiveCtx | 7.9 | 20.5 | 18.7 | 6.9 | 14.0 | – |
| LLMLingua | 7.3 | 19.6 | 16.3 | 6.3 | 14.2 | – |
| *Ours* | | | | | | |
| **CurvPrune-Pure** | 14.6 | 20.6 | 14.8 | 7.3 | 14.7 | – |
| **CurvPrune-Hybrid** | 15.0 | 20.6 | **32.5** | **8.5** | **15.2** | **+38%** |
| **BM25+Texture** | 18.2 | 20.3 | 38.4 | 7.9 | 14.9 | **+65%** |

■ best per column, ■ second-best. $\Delta\%$ = mean rel. F1 gain on multi-hop QA (HpQA, 2Wiki) vs. BM25. Means over $N{=}50$ examples.

*Table 2.* **Routing on multi-hop QA.** F1 and tokens at cost target $T$. +Tex. = Texture trigger.

| Method | HotpotQA | | 2WikiMQA | | $\Delta\%$ Tex. |
|---|---|---|---|---|---|
| | F1 | Tokens | F1 | Tokens | |
| *Adaptive baselines* | | | | | |
| FLARE | 39.1 | 175 | 38.2 | 182 | – |
| Self-Route | 45.4 | 7096 | 37.7 | 5780 | *ref* |
| Fixed-$k$ | 32.4 | 213 | 22.2 | 241 | – |
| *+Texture trigger* | | | | | |
| FLARE + Tex | 45.6 | **198** | **41.5** | **205** | **+13%** |
| Self-Route + Tex | **47.6** | 256 | 40.8 | 271 | **+7%** |
| *Ours* | | | | | |
| **CurvFlag** | **49.1** | 3344 | **43.0** | 3322 | **+11%** |

■ best (↑F1, ↓tokens), ■ second-best. $\Delta\%$ = relative F1 gain vs. matched baseline (FLARE / Self-Route; CurvFlag vs. Self-Route).

> **Curvature is actionable.** Texture delivers consistent multi-hop QA gains ($+10.3/+8.8$ F1 from BM25+Texture pruning, $+6.5/+3.3$ F1 from curvature-triggered routing), with CURVFLAG reaching higher F1 at $\sim2\times$ lower token cost than the strongest adaptive baseline.

of each curvature method over its matched baseline. Full budget sweeps ($B \in \{512, 1024, 2048, 4096\}$; Figure 5), per-pair deltas, cost–quality frontiers, and mechanism checks are in Appendix D.5.

**Analysis: curvature yields consistent gains by re-allocating budget toward bridge structure.** The clearest evidence that Texture captures task-relevant structure is the BM25→BM25+Texture pair: at $B{=}2048$ it improves BM25 by $+65\%$ relative F1 on multi-hop QA (Table 1) while staying within noise on the three non-multi-hop tasks. CURVPRUNE-PURE, the curvature-only ablation, isolates this signal: it is competitive on 2WikiMQA but trails query-aware selection on HotpotQA, since curvature locates *where* evidence is dispersed but not *which* content answers the query—hence the deployed methods combine it with BM25. Together, BM25+Texture and CURVPRUNE-HYBRID form the strongest compression frontier: BM25+Texture dominates multi-hop QA, while the hybrid variant is the best pruning method on Qasper and QMSum and ties for best on GovReport. On routing, CURVFLAG attains the best accuracy of any method while using $\sim2\times$ fewer tokens than Self-Route, so it *Pareto-dominates* the strongest adaptive baseline; the curvature-triggered FLARE+Texture and Self-Route+Texture variants are likewise positive at a fraction of the token cost (Table 2). In both, curvature re-allocation helps most where bridging evidence is dispersed (multi-hop QA) and is neutral where a single lexical anchor suffices.

**Transversal instantiation.** A relational instantiation over dependency/OpenIE graphs yields a curvature only weakly correlated with the longitudinal signal (Spearman $\rho{\approx}0.20$ on 200 WikiText-2 slots)—complementary, not redundant, structure; we report this rank agreement and the extractor cost in Appendix D.8.

## 8. Conclusion

We presented a unified framework for *text curvature* along three axes: definition-independent certificates that two-sided inference over natural text is non-flat (Section 4); *Texture*, a signed operator reading how two-sided reconciliation contracts (*focus*) or expands (*fan-out*) over a context rectangle under a neutral transport geometry (Section 5); and training-free, generator-agnostic controllers for compression and retrieval routing (Sections 6–7).

**Scope and limitations.** Texture is extractor-dependent in magnitude, so we treat the scale of $\kappa$ as calibrated to the chosen oracle; non-flatness and sign are stable across five extractor checkpoints, support sizes, temperatures, and radii (Appendices D.6, E.5). Gains are strongest on multi-hop QA, where curvature re-allocates a relevance-selected budget toward dispersed bridging evidence. Promising directions include heat-kernel limits to continuous Ricci curvature and curvature-guided generation.

## Impact Statement

**Potential benefits.** This work proposes a text-native curvature measurement and demonstrates its use as an inference-time control signal. If validated at scale, curvature-guided compression and retrieval routing could reduce compute and latency in LLM systems, improving accessibility and lowering environmental and monetary costs. Curvature-based diagnostics may also improve interpretability by highlighting structurally pivotal regions in prompts and documents.

**Potential risks and misuse.** Curvature-guided selection can amplify biases present in the belief extractor or in the underlying corpora: if a model systematically assigns fan-out/focus in biased ways, retrieval and compression decisions may preferentially preserve or discard certain viewpoints. RAG systems may inadvertently retrieve harmful or misleading content if retrieval is triggered in sensitive fan-out regions. We mitigate these risks by (i) reporting bias-sensitive analyses when feasible, (ii) recommending conservative thresholds and human oversight for high-stakes domains, and (iii) isolating failure modes where curvature signals become unreliable. **Data and privacy.** Our experiments use public datasets (WikiText-2, OpenWebText, the LongBench task suite) and do not require collecting new personal data. All evaluated language models are publicly available checkpoints; we report no new training of generators.

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

# Appendix

## Contents of the Appendix

## A. Notation Reference

Table 3 summarizes notation used throughout; detailed definitions appear in the indicated sections.

*Table 3.* **Notation.** Summary of symbols used throughout the paper.

| Symbol | Description | Ref. |
|---|---|---|
| *Text, Slots, and Beliefs* | | |
| $x_{1:n}, x_{<i}, x_{>i}$ | Token sequence; prefix; suffix | §3 |
| $x_{<i}^L, x_{>i}^R$ | Truncated contexts of radii $L, R$ | §3 |
| $(x_{<i}, \square, x_{>i})$ | Contextual slot at position $i$ | §3 |
| $\mathcal{S}_i, \mathcal{C}_i, \text{tail}$ | State space; candidate set; tail bucket | §3 |
| $\Delta(\mathcal{S}), \Delta^\circ(\mathcal{S})$ | Simplex; interior (full support) | §3 |
| $\mu_i^L, \mu_i^R$ | Left/right boundary beliefs | §5 |
| $\mu_i(L, R)$ | Two-sided posterior at radii $(L, R)$ | §4 |
| $\mathcal{B}_{\leftrightarrow}$ | Two-sided belief extractor | §3 |
| *Existence Certificates (Part I)* | | |
| $\text{KL}(\mu\|\nu)$ | Kullback–Leibler divergence | §3 |
| $u_{i,s}(L, R)$ | Log-odds: $\log[\mu_i(L,R)(s)/\mu_i(L,R)(s_{\text{ref}})]$ | §4 |
| $\Omega_{i,s}(L, R), h_i$ | Unit-square holonomy; aggregated magnitude | §4 |
| $\mu_i^{\leftarrow}, \mu_i^{\rightarrow}, \mu_i^{(0)}$ | Left-only, right-only, base beliefs | §4 |
| $\mu_i^{\text{PoE}}, \text{CEI}_i$ | Product-of-Experts; contextual evidence interaction | §4 |
| *Neutral Kernel and Dynamics (Part II)* | | |
| $c_i(s, s'), \varepsilon$ | Symmetric ground cost; temperature | §5 |
| $G_i, K_i, \pi_i$ | Gibbs affinity; Markov kernel; stationary dist. | §5 |
| $R_i(s_0, s_1, s_2)$ | Reference path: $\pi_i(s_0)K_i(s_0, s_1)K_i(s_1, s_2)$ | §5 |
| *Transport Divergence and Curvature (Part II)* | | |
| $C_\varepsilon(\rho, \sigma)$ | Entropic optimal-transport cost under $c_i$ | §5 |
| $\text{d}_i(\rho, \sigma)$ | Debiased Sinkhorn divergence (transport dissimilarity) | §5 |
| $(a, b), \gamma^\star$ | Sinkhorn scaling vectors; optimal coupling | App. C.2 |
| $\mu_{00}, \mu_{10}, \mu_{01}, \mu_{11}$ | Corners of a $2\times 2$ context rectangle | §5 |
| $\ell_L^0, \ell_L^R, \ell_R^0, \ell_R^L$ | Rectangle side lengths under $\text{d}_i$ | §5 |
| $s_i(g), \eta$ | Signed cell contraction score; ratio guard | §5 |
| $a_i(g), (\lambda, \tau_a)$ | Activity gate (Part I certificates); gate params | §5 |
| $\kappa_i^{\text{Tex}}, \epsilon_0$ | **Texture curvature** $\in [-1, 1]$; numerical guard | §5 |
| *Utility Controllers (Part III)* | | |
| $B, k$ | Token budget; retrieval budget | §6 |
| $s(I), (w_-, w_+)$ | Span score; fan-out/focus weights | App. D |
| $M_-(z)$ | Fan-out mass: $\sum_i [-\kappa_i^{\text{Tex}}]_+$ | App. D |

**Conventions.** Subscript $i$ denotes slot index. We write $\propto$ for equality up to normalization and $[x]_+ := \max(x, 0)$ for the positive part. When clear from context, slot subscripts may be suppressed.

## B. Additional Theory for Section 4

This appendix provides the full statements and proofs of the two *definition-independent* certificates used in Section 4. Both results are phrased as falsifiable *flatness nulls* on the two-sided posterior field $(L, R) \mapsto \mu_i(L, R)$. Throughout, we assume full support ($\mu_i(L, R) \in \Delta^\circ(\mathcal{S}_i)$); empirically this is ensured by a tail bucket and/or an $\varepsilon$-smoothing.

### B.1. Certificate I: Holonomy and Discrete Integrability of Log-Odds

**Theorem B.1** (Holonomy characterization of log-odds separability)**.** Curvature certificate (falsifiable flatness null). *Define the unit-square holonomy $\Omega_{i,s}(L, R)$ from the two-sided belief field. If there exist a state $s \in \mathcal{S}_i$ and an adjacent grid cell $(L, R)$ such that $\Omega_{i,s}(L, R) \neq 0$, then the flatness null fails: the log-odds field is* not *additively separable in $(L, R)$, hence the two-sided inference geometry exhibits genuine left–right interaction.*

**Setup.** *Fix a slot $i$ and a finite candidate/state set $\mathcal{S}_i$. Let $L_{\max}, R_{\max} \in \mathbb{N}$ and define the discrete grid domain*

$$\mathcal{D} := \{0, 1, \ldots, L_{\max}\} \times \{0, 1, \ldots, R_{\max}\}.$$

*(Only adjacency on the grid matters; if one uses nonuniform radii in practice, re-index them by their grid positions.) Assume a strictly positive belief field $\mu_i(L, R) \in \Delta^\circ(\mathcal{S}_i)$ for each $(L, R) \in \mathcal{D}$.*

*Fix a reference state $s_{\mathrm{ref}} \in \mathcal{S}_i$ and define the log-odds field*

$$u_{i,s}(L, R) := \log \frac{\mu_i(L, R)(s)}{\mu_i(L, R)(s_{\mathrm{ref}})} \qquad (s \in \mathcal{S}_i, \ (L, R) \in \mathcal{D}). \tag{14}$$

*Define the unit-square holonomy for $0 \le L < L_{\max}$ and $0 \le R < R_{\max}$ by*

$$\Omega_{i,s}(L, R) := u_{i,s}(L+1, R+1) - u_{i,s}(L+1, R) - u_{i,s}(L, R+1) + u_{i,s}(L, R). \tag{15}$$

**Claim (equivalence).** *The following are equivalent:*

$$\Omega_{i,s}(L, R) = 0 \ \text{ for all } s \in \mathcal{S}_i \text{ and all unit squares } (L, R), \tag{16}$$

*and for every $s \in \mathcal{S}_i$ there exist functions $\alpha_{i,s} : \{0, \ldots, L_{\max}\} \to \mathbb{R}$ and $\beta_{i,s} : \{0, \ldots, R_{\max}\} \to \mathbb{R}$ with $\alpha_{i,s}(0) = \beta_{i,s}(0) = 0$ such that*

$$u_{i,s}(L, R) = u_{i,s}(0, 0) + \alpha_{i,s}(L) + \beta_{i,s}(R) \qquad \text{for all } (L, R) \in \mathcal{D}. \tag{17}$$

**Discrete Stokes / telescoping identity.** *For any rectangle $0 \le L_1 < L_2 \le L_{\max}$ and $0 \le R_1 < R_2 \le R_{\max}$, define the rectangle holonomy*

$$\mathcal{H}_{i,s}(L_1, L_2, R_1, R_2) := u_{i,s}(L_2, R_2) - u_{i,s}(L_2, R_1) - u_{i,s}(L_1, R_2) + u_{i,s}(L_1, R_1). \tag{18}$$

*Then $\mathcal{H}_{i,s}$ decomposes into the sum of unit-square holonomies:*

$$\mathcal{H}_{i,s}(L_1, L_2, R_1, R_2) = \sum_{\ell=L_1}^{L_2-1} \sum_{r=R_1}^{R_2-1} \Omega_{i,s}(\ell, r). \tag{19}$$

*Proof.* We first prove the telescoping identity (19), then prove the equivalence between the flatness null (16) and the additive decomposition (17).

**Step 1: telescoping over a rectangle (proof of (19)).** Fix $s \in \mathcal{S}_i$ and a rectangle $(L_1, L_2, R_1, R_2)$. Sum (15) over $r = R_1, \ldots, R_2 - 1$ for a fixed $\ell$:

$$\sum_{r=R_1}^{R_2-1} \Omega_{i,s}(\ell, r) = \sum_{r=R_1}^{R_2-1} \Big( u_{i,s}(\ell+1, r+1) - u_{i,s}(\ell+1, r) \Big) - \sum_{r=R_1}^{R_2-1} \Big( u_{i,s}(\ell, r+1) - u_{i,s}(\ell, r) \Big)$$
$$= \big( u_{i,s}(\ell+1, R_2) - u_{i,s}(\ell+1, R_1) \big) - \big( u_{i,s}(\ell, R_2) - u_{i,s}(\ell, R_1) \big),$$

because each sum telescopes. Now sum this identity over $\ell = L_1, \ldots, L_2 - 1$; telescoping again yields

$$\sum_{\ell=L_1}^{L_2-1} \sum_{r=R_1}^{R_2-1} \Omega_{i,s}(\ell, r) = u_{i,s}(L_2, R_2) - u_{i,s}(L_2, R_1) - u_{i,s}(L_1, R_2) + u_{i,s}(L_1, R_1) = \mathcal{H}_{i,s}(L_1, L_2, R_1, R_2),$$

which is (19).

**Step 2: additive decomposition $\Rightarrow$ zero holonomy.** Assume (17). Substituting it into (15) shows all terms cancel:

$$\Omega_{i,s}(L, R) = \big( u_{i,s}(0, 0) + \alpha_{i,s}(L+1) + \beta_{i,s}(R+1) \big) - \big( u_{i,s}(0, 0) + \alpha_{i,s}(L+1) + \beta_{i,s}(R) \big)$$
$$- \big( u_{i,s}(0, 0) + \alpha_{i,s}(L) + \beta_{i,s}(R+1) \big) + \big( u_{i,s}(0, 0) + \alpha_{i,s}(L) + \beta_{i,s}(R) \big) = 0.$$

Thus (16) holds.

**Step 3: zero holonomy $\Rightarrow$ additive decomposition.** Assume (16). By Step 1, $\Omega \equiv 0$ implies $\mathcal{H} \equiv 0$ on every rectangle. In particular, for the rectangle with corners $(0, 0)$, $(L, 0)$, $(0, R)$, $(L, R)$ we obtain

$$0 = \mathcal{H}_{i,s}(0, L, 0, R) = u_{i,s}(L, R) - u_{i,s}(L, 0) - u_{i,s}(0, R) + u_{i,s}(0, 0),$$

hence

$$u_{i,s}(L, R) = u_{i,s}(L, 0) + u_{i,s}(0, R) - u_{i,s}(0, 0). \tag{20}$$

Define

$$\alpha_{i,s}(L) := u_{i,s}(L, 0) - u_{i,s}(0, 0), \qquad \beta_{i,s}(R) := u_{i,s}(0, R) - u_{i,s}(0, 0), \tag{21}$$

so $\alpha_{i,s}(0) = \beta_{i,s}(0) = 0$. Substituting (21) into (20) yields exactly (17).

**Step 4: certificate interpretation.** Steps 2–3 show that (16) is equivalent to additive separability (17). Therefore if any unit-square holonomy is nonzero, the additive decomposition cannot hold, which is precisely the stated curvature certificate. $\qquad\square$

**Remark (reference-state / gauge invariance).** Changing the reference state in (14) applies a gauge transformation to the log-odds coordinates: if $u'_{i,s}(L, R) = \log \frac{\mu_i(L,R)(s)}{\mu_i(L,R)(s'_{\mathrm{ref}})}$, then $u'_{i,s}(L, R) = u_{i,s}(L, R) - u_{i,s'_{\mathrm{ref}}}(L, R)$. Taking the mixed difference (15) gives $\Omega'_{i,s}(L, R) = \Omega_{i,s}(L, R) - \Omega_{i,s'_{\mathrm{ref}}}(L, R)$. Consequently, the *flatness null* "$\Omega_{i,s}(L, R) = 0$ for all $s$ and all unit squares" is invariant to the choice of reference. Moreover, the curvature certificate is also invariant: if some $\Omega_{i,s}$ is nonzero under one reference, then not all $\Omega'_{i,s}$ can vanish under another reference because $\Omega'_{i,s'_{\mathrm{ref}}} \equiv 0$ by construction.

## B.2. Certificate II: Evidence Additivity and Product-of-Experts (CEI)

**Theorem B.2** (Product-of-Experts characterization of evidence additivity). *Curvature certificate (falsifiable flatness null). Fix $(L, R)$. Define the PoE reconstruction $\mu_i^{\mathrm{PoE}}(L, R)$ from the boundary beliefs $\mu_i(L, 0)$ and $\mu_i(0, R)$ and the base belief $\mu_i(0, 0)$. If $\mu_i(L, R) \neq \mu_i^{\mathrm{PoE}}(L, R)$ (equivalently $\mathrm{CEI}_i(L, R) > 0$), then the additive-evidence flatness null fails: left and right context interact beyond additive log Bayes factors.*

Setup. *Fix a slot $i$ and a finite candidate/state set $\mathcal{S}_i$. Assume a strictly positive two-sided belief field $\mu_i(L, R) \in \Delta^\circ(\mathcal{S}_i)$. For a fixed pair of radii $(L, R)$ define the one-sided boundary beliefs*

$$\mu_i^\leftarrow(L) := \mu_i(L, 0), \qquad \mu_i^\rightarrow(R) := \mu_i(0, R), \qquad \mu_i^{(0)} := \mu_i(0, 0).$$

*Define the (normalized) product-of-experts reconstruction by*

$$\mu_i^{\mathrm{PoE}}(L, R)(s) := \frac{1}{Z_i(L, R)} \frac{\mu_i^\leftarrow(L)(s)\, \mu_i^\rightarrow(R)(s)}{\mu_i^{(0)}(s)}, \quad Z_i(L, R) := \sum_{t \in \mathcal{S}_i} \frac{\mu_i^\leftarrow(L)(t)\, \mu_i^\rightarrow(R)(t)}{\mu_i^{(0)}(t)}. \tag{22}$$

*Define the Contextual Evidence Interaction statistic*

$$\mathrm{CEI}_i(L, R) := \mathrm{KL}\Big(\mu_i(L, R) \,\big\|\, \mu_i^{\mathrm{PoE}}(L, R)\Big) \geq 0. \tag{23}$$

Claim 1 (PoE-flatness null $\Longleftrightarrow$ additive Bayes factors). *The following are equivalent:*

$$\mu_i(L, R) = \mu_i^{\mathrm{PoE}}(L, R)$$
$$\Longleftrightarrow \quad \exists\, c(L, R) \in \mathbb{R} \ \text{s.t.}$$
$$\log \mu_i(L, R)(s) = \log \mu_i^\leftarrow(L)(s) + \log \mu_i^\rightarrow(R)(s) - \log \mu_i^{(0)}(s) + c(L, R)$$

*for all $s \in \mathcal{S}_i$. Consequently,*

$$\mathrm{CEI}_i(L, R) = 0 \quad \Longleftrightarrow \quad \mu_i(L, R) = \mu_i^{\mathrm{PoE}}(L, R). \tag{24}$$

Claim 2 (variational characterization). *Define, for $q \in \Delta(\mathcal{S}_i)$,*

$$\mathcal{J}_{i,L,R}(q) := \mathrm{KL}(q \| \mu_i^\leftarrow) + \mathrm{KL}(q \| \mu_i^\rightarrow) - \mathrm{KL}(q \| \mu_i^{(0)}). \tag{25}$$

*Then $\mathcal{J}_{i,L,R}$ admits the exact identity*

$$\mathcal{J}_{i,L,R}(q) = \mathrm{KL}\Big(q \,\big\|\, \mu_i^{\mathrm{PoE}}(L,R)\Big) - \log Z_i(L,R), \tag{26}$$

*so $\mu_i^{\mathrm{PoE}}(L,R)$ is the unique minimizer of $\mathcal{J}_{i,L,R}$ and*

$$\mathrm{CEI}_i(L,R) = \mathcal{J}_{i,L,R}\big(\mu_i(L,R)\big) - \min_{q \in \Delta(\mathcal{S}_i)} \mathcal{J}_{i,L,R}(q). \tag{27}$$

**Claim 3 (generative sufficiency).** *Let $Z \in \mathcal{S}_i$ be the slot value and let $X^L, X^R$ denote the left/right context observations at radii $(L,R)$. If the beliefs correspond to exact posteriors $\mu_i(L,R)(\cdot) = \mathbb{P}(Z = \cdot \mid X^L, X^R)$, $\mu_i^{\leftarrow}(\cdot) = \mathbb{P}(Z = \cdot \mid X^{\leftarrow})$, $\mu_i^{\rightarrow}(\cdot) = \mathbb{P}(Z = \cdot \mid X^{\rightarrow})$, $\mu_i^{(0)}(\cdot) = \mathbb{P}(Z = \cdot)$, and if $X^{\leftarrow} \perp X^{\rightarrow} \mid Z$, then $\mu_i(L,R) = \mu_i^{\mathrm{PoE}}(L,R)$ for every realization with positive probability.*

*Proof.* **Step 1: Claim 1 (PoE $\Leftrightarrow$ additive Bayes factors).** Assume there exists $c(L,R)$ such that

$$\log \mu_i(L,R)(s) = \log \mu_i^{\leftarrow}(L)(s) + \log \mu_i^{\rightarrow}(R)(s) - \log \mu_i^{(0)}(s) + c(L,R) \quad \text{for all } s.$$

Exponentiating gives

$$\mu_i(L,R)(s) = e^{c(L,R)} \frac{\mu_i^{\leftarrow}(L)(s)\, \mu_i^{\rightarrow}(R)(s)}{\mu_i^{(0)}(s)}.$$

Summing both sides over $s$ and using $\sum_s \mu_i(L,R)(s) = 1$ yields $e^{c(L,R)} = 1/Z_i(L,R)$, hence $\mu_i(L,R) = \mu_i^{\mathrm{PoE}}(L,R)$.

Conversely, if $\mu_i(L,R) = \mu_i^{\mathrm{PoE}}(L,R)$, then by (22)

$$\mu_i(L,R)(s) = \frac{1}{Z_i(L,R)} \frac{\mu_i^{\leftarrow}(L)(s)\, \mu_i^{\rightarrow}(R)(s)}{\mu_i^{(0)}(s)},$$

and taking logs gives the additive identity with $c(L,R) = -\log Z_i(L,R)$. This proves Claim 1 and the zero-iff statement (24) follows from the basic property $\mathrm{KL}(P\|Q) = 0 \Leftrightarrow P = Q$.

**Step 2: Claim 2 (variational identity).** Write $\mu^{\leftarrow} := \mu_i^{\leftarrow}(L)$, $\mu^{\rightarrow} := \mu_i^{\rightarrow}(R)$, $\mu^{(0)} := \mu_i^{(0)}$ and abbreviate $\mu^{\mathrm{PoE}} := \mu_i^{\mathrm{PoE}}(L,R)$, $Z := Z_i(L,R)$. Expand (25) using $\mathrm{KL}(q\|p) = \sum_s q(s) \log \frac{q(s)}{p(s)}$:

$$\begin{aligned}
\mathcal{J}_{i,L,R}(q) &= \sum_s q(s) \log \frac{q(s)}{\mu^{\leftarrow}(s)} + \sum_s q(s) \log \frac{q(s)}{\mu^{\rightarrow}(s)} - \sum_s q(s) \log \frac{q(s)}{\mu^{(0)}(s)} \\
&= \sum_s q(s) \log q(s) - \sum_s q(s) \big( \log \mu^{\leftarrow}(s) + \log \mu^{\rightarrow}(s) - \log \mu^{(0)}(s) \big).
\end{aligned}$$

Define the unnormalized PoE density $\widetilde{\mu}^{\mathrm{PoE}}(s) := \mu^{\leftarrow}(s)\mu^{\rightarrow}(s)/\mu^{(0)}(s)$ so that $\mu^{\mathrm{PoE}}(s) = \widetilde{\mu}^{\mathrm{PoE}}(s)/Z$ and hence $\log \mu^{\mathrm{PoE}}(s) = \log \widetilde{\mu}^{\mathrm{PoE}}(s) - \log Z$. Then

$$\begin{aligned}
\mathrm{KL}(q\|\mu^{\mathrm{PoE}}) &= \sum_s q(s) \log \frac{q(s)}{\mu^{\mathrm{PoE}}(s)} = \sum_s q(s) \log q(s) - \sum_s q(s) \log \mu^{\mathrm{PoE}}(s) \\
&= \sum_s q(s) \log q(s) - \sum_s q(s) \log \widetilde{\mu}^{\mathrm{PoE}}(s) + \log Z = \mathcal{J}_{i,L,R}(q) + \log Z,
\end{aligned}$$

which rearranges to (26).

Because $\mu^{\mathrm{PoE}}$ has full support, $\mathrm{KL}(q\|\mu^{\mathrm{PoE}})$ is strictly convex in $q$ on the simplex. By (26), $\mathcal{J}_{i,L,R}$ differs only by an additive constant, so it is strictly convex as well and has a unique minimizer. The unique minimizer is achieved when $\mathrm{KL}(q\|\mu^{\mathrm{PoE}}) = 0$, i.e., $q = \mu^{\mathrm{PoE}}$. Finally, plugging $q = \mu_i(L,R)$ and $q^\star = \mu^{\mathrm{PoE}}$ into (26) yields (27).

**Step 3: Claim 3 (conditional independence sufficiency).** Assume $X^L \perp X^R \mid Z$ and fix any realization $(x^L, x^R)$ with positive probability. Bayes' rule gives

$$\mathbb{P}(Z = z \mid x^L, x^R) \propto \mathbb{P}(x^L, x^R \mid Z = z) \, \mathbb{P}(Z = z) = \mathbb{P}(x^L \mid Z = z) \, \mathbb{P}(x^R \mid Z = z) \, \mathbb{P}(Z = z),$$

where the equality uses conditional independence. Rewrite each likelihood term via Bayes:

$$\mathbb{P}(x^L \mid Z = z) = \frac{\mathbb{P}(Z = z \mid x^L)\,\mathbb{P}(x^L)}{\mathbb{P}(Z = z)}, \qquad \mathbb{P}(x^R \mid Z = z) = \frac{\mathbb{P}(Z = z \mid x^R)\,\mathbb{P}(x^R)}{\mathbb{P}(Z = z)}.$$

Substituting and absorbing factors independent of $z$ into the proportionality constant yields

$$\mathbb{P}(Z = z \mid x^L, x^R) \propto \frac{\mathbb{P}(Z = z \mid x^L)\,\mathbb{P}(Z = z \mid x^R)}{\mathbb{P}(Z = z)}.$$

Renormalizing over $z \in \mathcal{S}_i$ gives exactly the PoE form (22), hence $\mu_i(L, R) = \mu_i^{\mathrm{PoE}}(L, R)$ and $\mathrm{CEI}_i(L, R) = 0$. $\qquad\square$

**Remark (information-theoretic interpretation).**  Claim 3 identifies a concrete *generative* flatness null. Under misspecification, systematic nonzero CEI can be interpreted as a posterior-level signature of conditional dependence between the left and right contexts given the slot variable. The main text does not invoke this interpretation; the certificate is purely a statement about the belief field $\mu_i(L, R)$.

### B.3. Empirical Setup Details for Section 4

We instantiate the two-sided belief extractor $\mathcal{B}_{\leftrightarrow}$ with the frozen infilling model `distilroberta-base` (a distilled RoBERTa model; Liu et al., 2019; Sanh et al., 2019). We evaluate two corpora: WikiText-2 (raw) (Merity et al., 2016) and OpenWebText (10k documents) (Gokaslan & Cohen, 2019). For each corpus we sample 1000 slot positions $i$ uniformly from the valid range (excluding boundaries where needed for the largest radii), and evaluate beliefs on the radius grid $L, R \in \{0, 1, 2, 4, 8\}$.

**Slot state spaces and smoothing.**  For each slot we construct a fixed finite support $\mathcal{S}_i$ using top-$k$ candidates from the one-sided boundary beliefs (union of left/right top-$k$) plus a tail bucket (as in §3); all beliefs are projected onto $\mathcal{S}_i$ with residual mass assigned to tail. To satisfy the full-support assumptions of Appendix B.1–B.2, we apply a small $\varepsilon$-smoothing when necessary.

**Reference state and gauge-free summary.**  We pin $s_{\mathrm{ref}} := \arg\max_{s \in \mathcal{S}_i} \mu_i(0, 0)(s)$ per slot (the most-probable state under the base belief; deterministic and per-slot stable). The flatness null $\Omega_{i,s} \equiv 0$ is gauge-invariant w.r.t. the choice of $s_{\mathrm{ref}}$, but the empirical magnitude $h_i$ is not; we therefore additionally report the gauge-free pairwise summary

$$h_i^{\mathrm{pair}} := \left( \frac{1}{|\mathcal{G}|} \sum_{(L,R)\in\mathcal{G}} \sum_{s,t} w_s\, w_t\, [\Omega_{i,s}(L, R) - \Omega_{i,t}(L, R)]^2 / 2 \right)^{1/2},$$

which does not depend on any reference state because it uses pairwise differences $\Omega_s - \Omega_t$.

**Controls.**  We compare **Real** text to two coherence-destroying controls. **Suffix-swap** pairs sampled slots within a corpus and swaps their right-context windows, preserving marginal token statistics while breaking slot-specific two-sided coherence. **Local-shuffle** independently permutes tokens within each right-context window, preserving the window multiset while destroying local order and compositional cues. (We keep the left context unchanged in both controls.)

**Reported statistics.**  Holonomy uses unit-square differences on the ordered radius grid; we report the per-slot holonomy magnitude $h_i$ in (4) (with state weights $w_{i,s}(L, R)$ as defined in (4)). CEI is reported at the maximal radii $(L_{\max}, R_{\max}) = (8, 8)$. Appendix B.4 reports medians, effect sizes, and 95% bootstrap confidence intervals.

### B.4. Additional Empirical Diagnostics for Section 4

This subsection provides additional diagnostics and full numeric summaries for the existence certificates in Section 4; the experimental setup is in Appendix B.3. In the main paper we include only Panels (A–B); here we provide Panels (C–D) and tables with medians, effect sizes, and bootstrap confidence intervals. We report the per-slot holonomy magnitude $h_i$ from (4) and CEI at $(L_{\max}, R_{\max}) = (8, 8)$.

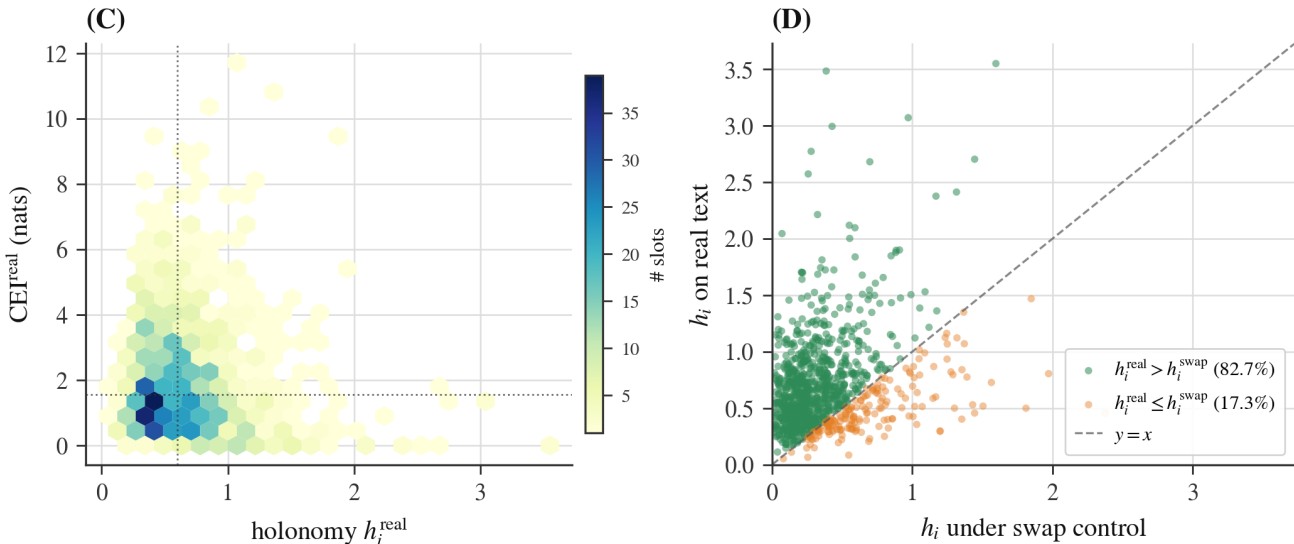

*Figure 4.* **Existence diagnostics (appendix: Panels C–D). (C)** Joint distribution of holonomy $h_i$ and CEI on real text (slot-level density): the two certificates are only loosely coupled, indicating that they capture complementary aspects of non-flatness rather than a single underlying statistic. **(D)** Slot-paired holonomy under real text versus the suffix-swap control: most points lie above the diagonal ($h_i^{\text{real}} > h_i^{\text{swap}}$), so coherent two-sided context induces stronger order-sensitive evidence interaction than the matched control.

*Table 4.* **Certificate I (Holonomy): summary statistics.** Medians over 1000 slots per dataset. $\Delta = \text{median}(\text{Real}) - \text{median}(\text{Control})$ with 95% bootstrap CIs.

| | **Median $h_i$** | | | **$\Delta$ [95% CI]** | |
|---|---|---|---|---|---|
| **Dataset** | **Real** | **Swap** | **Shuffle** | **Real$-$Swap** | **Real$-$Shuffle** |
| WikiText-2 | 0.596 | 0.305 | 0.373 | **0.291** [0.27, 0.31] | **0.223** [0.20, 0.25] |
| OpenWebText | 0.586 | 0.326 | 0.391 | **0.260** [0.24, 0.29] | **0.195** [0.18, 0.22] |

All Real$-$Control gaps are positive with 95% CIs excluding zero (bootstrap, $N{=}1000$).

# C. Additional Theory for Texture (Section 5)

This appendix supplies the technical foundations underlying the definition of Texture in Section 5. The main text defines Texture as a composition of three steps: (i) build a reversible neutral semantic-motion kernel from a ground cost and turn it into a debiased Sinkhorn transport distance $d_i$ between beliefs, (ii) evaluate the two-sided posterior field $\mu_i(L, R)$ on a context-radius grid, and (iii) read signed curvature as the activity-weighted transport contraction of that field across context rectangles. The goal here is to justify that the kernel/transport divergence is well-posed, computable, and stable, and to state the four defining properties of Contextual Ricci Texture.

Throughout this appendix we fix a slot $i$ and suppress the subscript $i$ to simplify notation. All objects below should be read as slot-local. We write $\mathcal{S}$ for the finite state space (e.g. $\mathcal{S} = \mathcal{C} \cup \{\text{tail}\}$ in the main text), and assume $\mu^L, \mu^R \in \Delta^\circ(\mathcal{S})$ unless stated otherwise.

## C.1. Neutral Kernel: Reversibility from a Symmetric Cost

The neutral semantic motion kernel in the main text is induced from a symmetric ground cost $c : \mathcal{S} \times \mathcal{S} \to \mathbb{R}_{\geq 0}$ via a Gibbs affinity $G(s, s') = \exp(-c(s, s')/\varepsilon)$ and row-normalization $K(s, s') = G(s, s')/\sum_u G(s, u)$. The following lemma records the key structural property: reversibility.

**Lemma C.1** (Reversibility of the Gibbs row-normalized kernel). *Assume $G(s, s') > 0$ and $G(s, s') = G(s', s)$ for all $s, s' \in \mathcal{S}$. Define*

$$K(s, s') := \frac{G(s, s')}{\sum_{u \in \mathcal{S}} G(s, u)}.$$

*Table 5.* **Certificate II (CEI): summary statistics.** Medians (nats) at $(L_{\max}, R_{\max}) = (8, 8)$ over 1000 slots. $\Delta = \text{median(Real)} - \text{median(Control)}$ with 95% bootstrap CIs.

| | **Median CEI (nats)** | | | **$\Delta$ [95% CI]** | |
|---|---|---|---|---|---|
| **Dataset** | **Real** | **Swap** | **Shuffle** | **Real$-$Swap** | **Real$-$Shuffle** |
| WikiText-2 | 1.514 | 0.697 | 1.319 | **0.816** [0.68, 0.95] | **0.195** [0.04, 0.33] |
| OpenWebText | 1.274 | 0.571 | 1.107 | **0.703** [0.54, 0.84] | **0.168** [0.01, 0.32] |

All Real$-$Control gaps are positive with 95% CIs excluding zero (bootstrap, $N=1000$).

*Let $\pi$ be defined (up to normalization) by*

$$\pi(s) \ \propto \ \sum_{u \in \mathcal{S}} G(s, u).$$

*Then $K$ is reversible with respect to $\pi$, i.e. $\pi(s)K(s, s') = \pi(s')K(s', s)$ for all $s, s'$. Consequently, $\pi$ is stationary: $\pi^\top K = \pi^\top$.*

*Proof.* By definition,

$$\pi(s)K(s, s') \ \propto \ \Big( \sum_u G(s, u) \Big) \frac{G(s, s')}{\sum_u G(s, u)} = G(s, s') = G(s', s) \ \propto \ \pi(s')K(s', s).$$

Summing the detailed balance identity over $s$ yields stationarity. $\qquad\square$

**Tail bucket geometry and full support.** The main text constructs a canonical slot support $\mathcal{S} = \mathcal{C} \cup \{\text{tail}\}$ to avoid renormalization artifacts from truncation. To prevent the tail state from becoming either an absorbing sink (too cheap) or an unreachable outlier (too expensive), a conservative choice is to set, for each $s \in \mathcal{C}$,

$$c(\text{tail}, s) = c(s, \text{tail}) := \text{median}_{u \in \mathcal{C}} \, c(u, s), \qquad c(\text{tail}, \text{tail}) = 0,$$

and then define $G, K$ from this augmented cost as in the main text. All theoretical statements below assume strictly positive kernels and interior beliefs; in implementation we enforce these mild conditions by tiny smoothing. For example, for $\delta \in (0, 1)$ one may replace boundary beliefs by $\tilde{\mu} = (1 - \delta)\mu + \delta \, \text{Unif}(\mathcal{S})$, and similarly add a vanishingly small constant to $G$ before row-normalization when needed.

**Why reversibility matters.** Reversibility is the algebraic form of left–right symmetry for the neutral reference dynamics. Starting the reference path measure from $\pi$ produces a time-reversible baseline, ensuring that the reconciliation step does not privilege either boundary belief by construction.

### C.2. Two-Step Schrödinger Bridge: Reduction and Scaling Form

As optional foundations for Texture's neutral transport geometry, we record the two-step Schrödinger bridge and its Sinkhorn scaling form. The canonical object used in the main text is the entropic optimal-transport cost $C_\varepsilon$ of (7) (and its debiased divergence $\text{d}_i$), computed directly by Sinkhorn iterations; the bridge below recovers the same coupling and is used only for the appendix diagnostics. Fix a strictly positive Markov kernel $K$ on $\mathcal{S}$ and let $\pi$ be any strictly positive stationary distribution for $K$ (in our construction, Lemma C.1 guarantees this). Define the two-step reference path measure on $\mathcal{S}^3$:

$$R(s_0, s_1, s_2) := \pi(s_0) \, K(s_0, s_1) \, K(s_1, s_2). \tag{28}$$

Given boundary beliefs $\mu^L, \mu^R \in \Delta^\circ(\mathcal{S})$, the two-step Schrödinger bridge is

$$P^\star \in \arg \min_{P: \, P_0 = \mu^L, \, P_2 = \mu^R} \text{KL}(P \| R), \tag{29}$$

and (recorded only as optional foundations, not used by the Texture operator) the midpoint is $\mu^{\text{mid}} := (P^\star)_1$.

**Theorem C.2** (Two-step bridge structure and computability). *Assume $K(s, s') > 0$ for all $s, s' \in \mathcal{S}$ and $\mu^L, \mu^R \in \Delta^\circ(\mathcal{S})$. Let $R$ be defined by* (28). *Then:*

**(i) Existence and uniqueness.**   *The bridge problem* (29) *admits a unique minimizer* $P^\star$.

**(ii) Endpoint reduction.**   *Let* $M := K^2$ *be the two-step kernel and define the endpoint reference coupling*

$$R_{02}(s_0, s_2) := \sum_{s_1} R(s_0, s_1, s_2) = \pi(s_0)\, M(s_0, s_2). \tag{30}$$

*Let* $\gamma^\star := (P^\star)_{02}$. *Then* $\gamma^\star$ *is the unique minimizer of the static KL projection*

$$\gamma^\star \in \arg\min_{\gamma:\ \gamma_0 = \mu^L,\ \gamma_2 = \mu^R} \mathrm{KL}(\gamma \| R_{02}), \tag{31}$$

*and* $P^\star$ *is obtained by pairing* $\gamma^\star$ *with the* reference *conditional of* $S_1$ *given endpoints:*

$$P^\star(s_0, s_1, s_2) = \gamma^\star(s_0, s_2)\, R(s_1 \mid s_0, s_2), \qquad R(s_1 \mid s_0, s_2) = \frac{K(s_0, s_1) K(s_1, s_2)}{M(s_0, s_2)}. \tag{32}$$

**(iii) Scaling form.**   *There exist positive vectors* $a, b : \mathcal{S} \to \mathbb{R}_{>0}$, *unique up to the gauge* $a \leftarrow ca$, $b \leftarrow c^{-1}b$, *such that*

$$\gamma^\star(s_0, s_2) = a(s_0)\, R_{02}(s_0, s_2)\, b(s_2), \tag{33}$$

*with* $(a, b)$ *satisfying the Schrödinger system*

$$\mu^L(s_0) = a(s_0) \sum_{s_2} R_{02}(s_0, s_2) b(s_2), \qquad \mu^R(s_2) = b(s_2) \sum_{s_0} a(s_0) R_{02}(s_0, s_2). \tag{34}$$

*In particular,* $(a, b)$ *can be computed by iterative proportional fitting (Sinkhorn/IPFP) on the strictly positive matrix* $R_{02}$.

**(iv) Midpoint message passing (optional foundations; not used by the Texture operator).**   *Let*

$$f(s_1) := \sum_{s_0} a(s_0)\pi(s_0)K(s_0, s_1), \qquad g(s_1) := \sum_{s_2} K(s_1, s_2)b(s_2).$$

*Then the midpoint marginal satisfies the factorization*

$$\mu^{\mathrm{mid}}(s_1) = f(s_1)\, g(s_1) \qquad \forall s_1 \in \mathcal{S}. \tag{35}$$

*Proof.*   We prove the four claims in order.

**(i) Existence and uniqueness.**   The feasible set

$$\mathcal{F} := \{ P \in \Delta(\mathcal{S}^3) : P_0 = \mu^L,\ P_2 = \mu^R \}$$

is a nonempty convex polytope (linear constraints intersected with a simplex), hence compact. Since $R(s_0, s_1, s_2) > 0$ for all triples, $\mathrm{KL}(P\|R)$ is finite on $\mathcal{F}$ and is strictly convex in $P$ because $x \mapsto x \log x$ is strictly convex on $\mathbb{R}_{\geq 0}$. A strictly convex lower-semicontinuous function on a compact convex set has a unique minimizer, so $P^\star$ exists and is unique.

**(ii) Endpoint reduction.**   Let $\gamma := P_{02}$ and let $Q(\cdot \mid s_0, s_2)$ denote the conditional distribution of $S_1$ given $(S_0, S_2) = (s_0, s_2)$ under $P$. Because $R$ is strictly positive, its corresponding conditional $R(\cdot \mid s_0, s_2)$ is well-defined and given by (32). The KL chain rule yields the decomposition

$$\mathrm{KL}(P\|R) = \mathrm{KL}(\gamma \| R_{02}) + \mathbb{E}_{(s_0, s_2) \sim \gamma}\Big[ \mathrm{KL}\big( Q(\cdot \mid s_0, s_2) \,\|\, R(\cdot \mid s_0, s_2) \big) \Big]. \tag{36}$$

The second term is nonnegative and equals zero iff $Q(\cdot \mid s_0, s_2) = R(\cdot \mid s_0, s_2)$ for $\gamma$-almost every $(s_0, s_2)$. Therefore, among all $P$ with a fixed endpoint marginal $\gamma$, the unique minimizer of $\mathrm{KL}(P\|R)$ is obtained by choosing the reference conditional at every endpoint pair, i.e. $P(s_0, s_1, s_2) = \gamma(s_0, s_2)R(s_1 \mid s_0, s_2)$. Thus, minimizing (29) reduces to minimizing the first term in (36), which is exactly the static problem (31). Since the static objective is strictly convex in $\gamma$, the minimizer $\gamma^\star$ is unique. Plugging $\gamma^\star$ back into the minimizing conditional gives (32) and hence the dynamic optimizer $P^\star$.

**(iii) Scaling form.** Consider the static problem (31). Form the Lagrangian with multipliers $\lambda : \mathcal{S} \to \mathbb{R}$ and $\nu : \mathcal{S} \to \mathbb{R}$ (distinct from the ratio guard $\eta$ used in the cell score):

$$\mathcal{L}(\gamma, \lambda, \nu) = \sum_{s_0, s_2} \gamma(s_0, s_2) \log \frac{\gamma(s_0, s_2)}{R_{02}(s_0, s_2)} + \sum_{s_0} \lambda(s_0) \Big( \mu^L(s_0) - \sum_{s_2} \gamma(s_0, s_2) \Big) + \sum_{s_2} \nu(s_2) \Big( \mu^R(s_2) - \sum_{s_0} \gamma(s_0, s_2) \Big).$$

At the (unique) optimum, first-order optimality holds and $\gamma^\star$ has full support (because $R_{02} > 0$ and $\mu^L, \mu^R > 0$), so differentiating w.r.t. $\gamma(s_0, s_2)$ gives

$$\log \frac{\gamma^\star(s_0, s_2)}{R_{02}(s_0, s_2)} + 1 - \lambda(s_0) - \nu(s_2) = 0.$$

Exponentiating yields

$$\gamma^\star(s_0, s_2) = R_{02}(s_0, s_2) \exp(\lambda(s_0) - 1) \exp(\nu(s_2)) =: a(s_0) R_{02}(s_0, s_2) b(s_2),$$

which is (33). The marginal constraints become exactly (34). Gauge freedom follows because multiplying $a$ by $c$ and dividing $b$ by $c$ leaves $\gamma^\star$ unchanged.

**(iv) Midpoint message passing.** Combine (32) with the scaling form (33). Using $R_{02}(s_0, s_2) = \pi(s_0) M(s_0, s_2)$ and $R(s_1 \mid s_0, s_2) = K(s_0, s_1) K(s_1, s_2) / M(s_0, s_2)$, we obtain

$$P^\star(s_0, s_1, s_2) = a(s_0) \pi(s_0) M(s_0, s_2) b(s_2) \cdot \frac{K(s_0, s_1) K(s_1, s_2)}{M(s_0, s_2)} = a(s_0) \pi(s_0) K(s_0, s_1) K(s_1, s_2) b(s_2).$$

Summing over $s_0, s_2$ yields

$$\mu^{\mathrm{mid}}(s_1) = \sum_{s_0, s_2} a(s_0) \pi(s_0) K(s_0, s_1) K(s_1, s_2) b(s_2) = \Big( \sum_{s_0} a(s_0) \pi(s_0) K(s_0, s_1) \Big) \Big( \sum_{s_2} K(s_1, s_2) b(s_2) \Big),$$

which is exactly (35). $\qquad\square$

**Implementation note (Sinkhorn/IPFP).** Writing $Q := R_{02}$ as a strictly positive $|\mathcal{S}| \times |\mathcal{S}|$ matrix, the Schrödinger system (34) is solved by iterative proportional fitting:

$$a^{(t+1)} = \mu^L \oslash (Q b^{(t)}), \qquad b^{(t+1)} = \mu^R \oslash (Q^\top a^{(t+1)}),$$

where $\oslash$ denotes elementwise division and $(Qb)(s_0) = \sum_{s_2} Q(s_0, s_2) b(s_2)$. On strictly positive matrices this converges to the scaling vectors yielding the unique $\gamma^\star$, and hence $\mu^{\mathrm{mid}}$ via (35). Standard references include Sinkhorn–Knopp and modern OT treatments (Cuturi, 2013).

### C.3. Properties of Contextual Ricci Texture

Contextual Ricci Texture (Section 5.3) reads signed curvature from how the two-sided posterior field $\mu_i(L, R)$ contracts or expands across a $2 \times 2$ context rectangle, measured by the debiased Sinkhorn divergence $\mathrm{d}_i$ (8). We collect its four defining properties; sign conventions and the activity-weighted aggregation (10) are as in the main text. Throughout, a cell $g$ has corners $\mu_{00}, \mu_{10}, \mu_{01}, \mu_{11}$ and side lengths $\ell_L^0, \ell_L^R, \ell_R^0, \ell_R^L \geq 0$, and $s_i(g)$ is its signed contraction (9).

**Proposition C.3** (Boundedness). *Fix guards $\eta > 0$, $\epsilon_0 > 0$ and activity-gate parameters $\lambda \geq 0$, $\tau_a > 0$. For every cell $g$, $s_i(g) \in [-1, 1]$, and therefore the activity-weighted and median aggregations satisfy $\kappa_i^{\mathrm{Tex}}, \kappa_i^{\mathrm{med}} \in [-1, 1]$.*

*Proof.* Each ratio $(\ell_L^0 - \ell_L^R) / (\ell_L^0 + \ell_L^R + \eta)$ has numerator bounded in absolute value by $\ell_L^0 + \ell_L^R$, hence by the denominator (as $\ell \geq 0$ and $\eta > 0$ make the denominator strictly positive), so it lies in $[-1, 1]$; $s_i(g)$ is the average of two such ratios, so $s_i(g) \in [-1, 1]$. The gate argument $(\|\Omega_i(g)\| + \lambda \, \mathrm{CEI}_i(g)) / \tau_a$ is nonnegative (a norm plus a nonnegative interaction term scaled by $\lambda \geq 0$, divided by $\tau_a > 0$), so the weights $a_i(g) = \tanh(\cdot) \in [0, 1)$ are nonnegative; hence $\kappa^{\mathrm{Tex}}$ is a convex combination of values in $[-1, 1]$ (the guard $\epsilon_0 > 0$ only shrinks magnitude), and the median of values in $[-1, 1]$ lies in $[-1, 1]$. $\qquad\square$

**Proposition C.4** (Flat-rectangle null). *If the field is* additively separable *on a cell, i.e. the transport increments satisfy* $\ell_L^0 = \ell_L^R$ *and* $\ell_R^0 = \ell_R^L$ *(the left increment is unaffected by right context and vice versa), then* $s_i(g) = 0$. *In particular, if* $\mu_i(L, R)$ *factors as an independent product-of-experts whose one-sided transport increments do not interact, every cell scores* $0$ *and* $\kappa_i^{\text{Tex}} = 0$.

*Proof.* Both signed ratios vanish identically when $\ell_L^0 = \ell_L^R$ and $\ell_R^0 = \ell_R^L$, giving $s_i(g) = 0$; aggregating zeros gives $\kappa^{\text{Tex}} = 0$. $\qquad\square$

**Proposition C.5** (Ricci-contraction interpretation). *Fix the left increment* $\delta_L$ *and view* $r \mapsto \ell_L(r) := d_i(\mu_i(L, r), \mu_i(L+\delta_L, r))$ *as the length of the left-increment "edge" transported along the right axis. Then* $s_i(g) > 0$ *iff this edge length* shrinks *as right context is added* ($\ell_L^R < \ell_L^0$) *on average with the symmetric right-edge term, i.e. iff parallel transport of evidence increments along one axis is contracted by motion along the other. This is the discrete two-axis analogue of positive Ricci curvature (nearby geodesics converge);* $s_i(g) < 0$ *is the diverging (fan-out) case.*

*Proof.* Immediate from (9): the first ratio is positive iff $\ell_L^R < \ell_L^0$, the second iff $\ell_R^L < \ell_R^0$, and $s_i(g)$ is their average; the geometric reading follows by identifying the four corners with the endpoints of two adjacent increment edges of the field. $\qquad\square$

**Remark (Ollivier-style reduction, as analogy).** Let $m_x, m_y$ be the one-step push-forwards of corners $x, y$ under the slot kernel $K_i$. The Ollivier–Ricci curvature $\kappa^{\text{Oll}}(x, y) = 1 - W_1(m_x, m_y)/d(x, y)$ compares a *transported* distance to a base distance along a single edge. Replacing $W_1$ by the debiased Sinkhorn divergence and the single edge $(x, y)$ by the two adjacent increment edges of a context rectangle recovers (9) up to the symmetric normalization $\ell + \ell' + \eta$. Thus $s_i(g)$ is a two-axis, rectangle-localized Ollivier–Ricci analogue; we use it as an interpretive analogy, not an identity (no single base point or geodesic is privileged). The correspondence is structural: both quantities are normalized differences of a transported length and a reference length, signed so that contraction is positive, and the normalization $\ell_L^0 + \ell_L^R + \eta$ replaces division by $d(x, y)$ while keeping the score bounded and symmetric in the two corners; we do not claim equality of constants.

**Caveat:** $\kappa_i^{\text{Tex}} = 0$ **does not certify full-field flatness.** The score aggregates local cell contractions; a field can have $\kappa_i^{\text{Tex}} \approx 0$ by cancellation of focusing and fan-out cells. Global non-flatness is the separate, definition-independent claim of Part I (holonomy/CEI), which is why the canonical aggregation *gates* cells by the Part I certificates $a_i(g)$: a slot is reported as strongly signed only when non-flatness is also certified.

**Aggregation ablation (median vs. activity-weighted).** We implement both $\kappa_i^{\text{Tex}}$ (activity-weighted) and $\kappa_i^{\text{med}} = \text{median}_g\, s_i(g)$. On WikiText-2 both are genuinely two-sided (each places substantial mass on positive and negative slots, with the activity-weighted form modestly fan-out-leaning); we select the activity-weighted form as canonical because it inherits its confidence from the same Part I certificates that establish curvature exists, and report the median variant as a robustness ablation.

## C.4. Stability and Truncation Consistency

Finally, we justify the stability of the bridge solution and the resulting curvature under perturbations of the inputs. This matters operationally because the finite state space $\mathcal{S}$ is induced by truncation (e.g. top-$k$ plus a tail bucket), and because beliefs and costs are computed numerically.

We emphasize that our setting is finite-state with strictly positive references and strictly positive endpoint marginals. In this interior regime, the bridge and its derived quantities vary continuously with the inputs. General stability results for entropic OT and Schrödinger bridges under weak convergence and cost perturbations have been studied in the OT literature.

**Theorem C.6** (Continuity of the bridge solution and Texture quantities). *Fix a finite state space* $\mathcal{S}$. *Let* $(\mu_n^L, \mu_n^R, K_n, \pi_n)$ *be a sequence of inputs such that:*

- $\mu_n^L, \mu_n^R \in \Delta^\circ(\mathcal{S})$ *for all* $n$ *and* $\mu_n^L \to \mu^L$, $\mu_n^R \to \mu^R$ *entrywise;*

- $K_n(s, s') > 0$ *for all* $s, s'$ *and* $K_n \to K$ *entrywise;*

- $\pi_n$ *is stationary for* $K_n$ *and* $\pi_n \to \pi$ *entrywise.*

*Let $R_n$ be the corresponding reference path measures, and let $P_n^\star$ be the unique bridge minimizers. Let $C_{\varepsilon,n}$, $\mathsf{d}_{i,n}$, $s_{i,n}(g)$, and $\kappa_n^{\mathrm{Tex}}$ be the corresponding entropic transport costs, debiased Sinkhorn divergences, cell contraction scores, and Texture curvature (defined as in Section 5, with the same guards $\eta, \epsilon_0$). Then*

$$P_n^\star \to P^\star, \quad C_{\varepsilon,n} \to C_\varepsilon, \quad \mathsf{d}_{i,n} \to \mathsf{d}_i, \quad s_{i,n}(g) \to s_i(g), \quad \kappa_n^{\mathrm{Tex}} \to \kappa^{\mathrm{Tex}}$$

*entrywise (and hence in total variation).*

*As a corollary, if $\mathcal{S}^{(k)}$ is an increasing sequence of truncation supports and the induced inputs $(\mu_{(k)}^L, \mu_{(k)}^R, K_{(k)}, \pi_{(k)})$ converge (under the canonical tail-bucket pushforward), then the computed Texture curvatures converge as $k \to \infty$.*

*Proof.* We sketch a self-contained finite-state argument.

**Step 1: reduce to the static problem.** By Theorem C.2, each dynamic bridge $P_n^\star$ is uniquely determined by its endpoint coupling $\gamma_n^\star = (P_n^\star)_{02}$, which uniquely solves the static KL projection

$$\gamma_n^\star \in \arg\min_{\gamma:\, \gamma_0 = \mu_n^L,\, \gamma_2 = \mu_n^R} \mathrm{KL}(\gamma \| R_{n,02}), \qquad R_{n,02}(s_0, s_2) = \pi_n(s_0)\,(K_n^2)(s_0, s_2).$$

Thus it suffices to show $\gamma_n^\star \to \gamma^\star$; the remaining claims follow by continuity of the explicit formulas.

**Step 2: compactness and identification of subsequential limits.** The set of couplings on $\mathcal{S} \times \mathcal{S}$ is a simplex and hence compact. Therefore $(\gamma_n^\star)$ has a convergent subsequence $\gamma_{n_j}^\star \to \bar\gamma$. Since taking marginals is linear and continuous, the limit coupling satisfies the limiting constraints: $\bar\gamma_0 = \mu^L$ and $\bar\gamma_2 = \mu^R$.

**Step 3: pass optimality to the limit.** Because $\mu^L, \mu^R$ are strictly positive and $R_{02}$ is strictly positive, the limiting optimizer $\gamma^\star$ has full support (by the scaling form), and the objective $\gamma \mapsto \mathrm{KL}(\gamma \| R_{02})$ is continuous in a neighborhood of $\gamma^\star$. Moreover, since $R_{n,02} \to R_{02}$ entrywise and $\gamma_{n_j}^\star \to \bar\gamma$, we have $\mathrm{KL}(\gamma_{n_j}^\star \| R_{n_j,02}) \to \mathrm{KL}(\bar\gamma \| R_{02})$ by finite-dimensional continuity.

Let $\gamma$ be any feasible coupling for the limit problem (i.e. $\gamma_0 = \mu^L$, $\gamma_2 = \mu^R$). For $j$ large enough, we can form a feasible coupling $\gamma^{(j)}$ for the $j$-th problem with marginals $(\mu_{n_j}^L, \mu_{n_j}^R)$ that converges to $\gamma$ as $j \to \infty$ (e.g. by a small amount of mass redistribution across rows/columns; this is always possible in finite dimension when all marginals are in the interior). Then optimality of $\gamma_{n_j}^\star$ yields

$$\mathrm{KL}(\gamma_{n_j}^\star \| R_{n_j,02}) \le \mathrm{KL}(\gamma^{(j)} \| R_{n_j,02}).$$

Taking $j \to \infty$ and using continuity gives $\mathrm{KL}(\bar\gamma \| R_{02}) \le \mathrm{KL}(\gamma \| R_{02})$ for all feasible $\gamma$. Hence $\bar\gamma$ is an optimizer for the limit problem; by uniqueness, $\bar\gamma = \gamma^\star$.

**Step 4: conclude full convergence and propagate to Texture quantities.** Since every subsequence has the same limit $\gamma^\star$, the whole sequence converges: $\gamma_n^\star \to \gamma^\star$. The explicit construction in Theorem C.2 then gives $P_n^\star \to P^\star$. The entropic transport cost $C_{\varepsilon,n}$ (7) is a continuous function of the optimal coupling and the marginals, hence $C_{\varepsilon,n} \to C_\varepsilon$; the debiased Sinkhorn divergence $\mathsf{d}_{i,n}$ (8) is a continuous (positive-part square-root) function of these costs, so $\mathsf{d}_{i,n} \to \mathsf{d}_i$. Each cell score $s_{i,n}(g)$ (9) is a continuous function of the four side lengths (the $\eta$ guard keeps every denominator bounded away from zero), so $s_{i,n}(g) \to s_i(g)$; and $\kappa_n^{\mathrm{Tex}} \to \kappa^{\mathrm{Tex}}$ by continuity of the activity-weighted average (the $\epsilon_0$ guard makes its denominator uniformly positive).

The truncation corollary follows by applying the same argument to the sequence of induced finite supports and noting that the tail-bucket pushforward produces interior marginals on each $\mathcal{S}^{(k)}$ and consistent convergence of inputs. $\qquad\square$

**Takeaway.** In the finite-state interior regime relevant to our computation (strictly positive kernels and beliefs), the transport divergences, cell contraction scores, and the resulting Texture curvature are stable functions of the inputs. This is the formal justification for computing Texture on truncated candidate sets and for interpreting it as a robust slot-local geometric quantity.

---

**Algorithm 1** TEXTURE($i$): Contextual Ricci Texture at slot $i$

---

**Require:** Slot index $i$; belief extractor for the posterior field $\mu_i(L, R)$; radius grid $\mathcal{G} = \{0, 1, 2, 4, 8, 16\}^2$; candidate budget $k$; symmetric cost builder $c_i$; temperature $\varepsilon$; ratio guard $\eta$; activity params $(\lambda, \tau_a)$; guard $\epsilon_0$.
**Ensure:** Curvature $\kappa_i^{\text{Tex}} \in [-1, 1]$ and (optionally) per-cell contraction traces.
1: Evaluate the field $\mu_i(L, R)$ on every grid node $(L, R) \in \mathcal{G}$ (one masked-LM forward per node, batched).
2: Form the canonical support $\mathcal{S}_i = \text{TopK}(\mu_i(L_{\max}, 0)) \cup \text{TopK}(\mu_i(0, R_{\max})) \cup \{\text{tail}\}$ and project every corner belief onto $\mathcal{S}_i$ (leftover mass $\to$ tail; tiny smoothing for full support).
3: Build a symmetric cost $c_i$ on $\mathcal{S}_i$ (with conservative tail geometry); normalize its scale.
4: **for** each adjacent $2 \times 2$ cell $g \in \mathcal{G}$ **do**
5:   Compute the four side lengths via the debiased Sinkhorn divergence (8) (log-domain Sinkhorn; memoize self-costs and shared-edge distances).
6:   Form the signed contraction $s_i(g) \in [-1, 1]$ via (9) (guard near-flat cells to 0).
7:   Form the activity gate $a_i(g) = \tanh((\|\Omega_i(g)\| + \lambda \, \text{CEI}_i(g))/\tau_a)$ from the corner log-odds holonomy and per-cell CEI.
8: **end for**
9: **Return** $\kappa_i^{\text{Tex}} \leftarrow \left( \sum_g a_i(g) s_i(g) \right) / \left( \sum_g a_i(g) + \epsilon_0 \right)$ from (10); optionally the median variant $\kappa_i^{\text{med}} = \text{median}_g \, s_i(g)$.

---

## C.5. Algorithmic Summary

For completeness we include a reference implementation of the per-slot computation. The only nontrivial step is solving the endpoint scaling system (Sinkhorn/IPFP) on the strictly positive matrix $R_{02}$ (Theorem C.2). On typical top-$k$ supports this is inexpensive; the complexity is dominated by repeated multiplications with the $|\mathcal{S}| \times |\mathcal{S}|$ matrix $R_{02}$.

## C.6. Transversal Texture: Curvature on Relational Graphs

The main text presents Texture as a single curvature operator that takes as input (i) a finite slot-local state space $\mathcal{S}_i$, (ii) two one-sided boundary beliefs $\mu_i^L, \mu_i^R \in \Delta(\mathcal{S}_i)$, and (iii) a symmetric geometry (or equivalently a reversible neutral motion kernel). In the longitudinal instantiation, the state space consists of *token-filler* hypotheses (top-$k$ candidates plus a tail bucket). This appendix defines a complementary *transversal* instantiation in which the state space consists of *relational* hypotheses induced from structured semantic graphs.

**Why transversal curvature.** Longitudinal curvature quantifies whether prefix and suffix evidence collapse into a common basin over candidate fillers at position $i$. Transversal curvature instead quantifies whether prefix and suffix evidence collapse into a common basin over *relational structure* surrounding the slot, e.g. which entity participates in which event, which argument role a mention fills, or which relation is supported. This captures structural ambiguity that may be invisible at the token-filler level.

### C.6.1. GRAPH SOURCES: AMR AND OPENIE (AND BEYOND)

Transversal Texture is designed to be *representation-agnostic*: any procedure that maps a sentence to a small relational graph can be used to instantiate $\mathcal{S}_i$ and its geometry.

**AMR graphs.** Abstract Meaning Representation (AMR) represents a sentence as a rooted, directed, edge-labeled semantic graph whose nodes are abstract concepts (often PropBank frames) and whose edges encode semantic roles and modifiers. AMR is intentionally closer to a meaning representation than surface syntax and is well-suited for defining predicate–argument neighborhoods. (Banarescu et al., 2013)

**OpenIE graphs.** Open Information Extraction (OpenIE) systems extract relation tuples from text in a schema-free fashion, typically producing triples of the form (*arg1*, *relation*, *arg2*). These triples can be interpreted as a lightweight sentence-level knowledge graph, and can also be aggregated across a corpus to form a larger KG. OpenIE is attractive for transversal Texture because it is scalable and directly yields relational alternatives. (Etzioni et al., 2008; Angeli et al., 2015)

**Relationship and complementarity.** AMR and OpenIE are both relational views of text, but they differ in canonicalization and noise profile: AMR typically yields a single compositional semantic graph with typed roles, while OpenIE yields a set of surface-attested relation tuples that are broad-coverage but less canonical. In transversal Texture, these differences become useful: AMR emphasizes structured event/role semantics, while OpenIE emphasizes extracted relational facts that can be pooled across data.

C.6.2. RELATIONAL STATE SPACE AT SLOT $i$

Fix a sentence $x_{1:n}$ and slot index $i$. Let $\mathsf{G}^{(m)}(x)$ be a graph extractor for source $m$ (e.g. AMR, OpenIE) returning a directed labeled graph $\mathcal{G}^{(m)} = (V^{(m)}, E^{(m)})$ along with a (possibly approximate) alignment map $\mathrm{anch}^{(m)} : V^{(m)} \to 2^{\{1,\dots,n\}}$ that associates each node with a token span.

We define a slot-local relational candidate set by restricting to a neighborhood of anchors around the slot. Let $\mathcal{A}_i$ be a small anchor set (e.g. tokens in a window around $i$, or tokens that have high attention to the slot). For each source $m$, define the node neighborhood

$$V_i^{(m)} := \{v \in V^{(m)} : \mathrm{anch}^{(m)}(v) \cap \mathcal{A}_i \neq \emptyset\} \cup \{u : \exists v \in V^{(m)} \text{ with } (u,v) \in E^{(m)} \text{ or } (v,u) \in E^{(m)}\},$$

i.e. anchors plus one-hop relational context. Finally, define the transversal state space as a tagged union across sources plus a tail bucket:

$$\mathcal{S}_i^{\perp} := \left( \bigsqcup_m V_i^{(m)} \right) \cup \{\mathrm{tail}\}. \tag{37}$$

The tag in the disjoint union is important: it prevents accidental identification of nodes across different extractors, but still permits controlled fusion via shared token anchors (below).

**Tail bucket.** The state $\mathrm{tail}$ absorbs residual mass for relational hypotheses not represented in the induced neighborhood, or failures of the graph extractor. This mirrors the longitudinal tail-bucket construction.

C.6.3. MULTI-GRAPH FUSION GEOMETRY

Transversal Texture requires a symmetric geometry on $\mathcal{S}_i^{\perp}$. We propose a simple, general *multi-graph fusion* recipe that produces a symmetric affinity matrix (and hence a reversible neutral kernel by Lemma C.1).

**Step 1: source-wise symmetric adjacency.** For each source $m$, build a weighted undirected adjacency $W^{(m)}$ on $\mathcal{S}_i^{\perp}$ as follows. If $(u,v) \in E^{(m)}$ (directed), add weight to the undirected edge between tagged nodes $(m,u)$ and $(m,v)$:

$$W^{(m)}\big((m,u),(m,v)\big) \leftarrow W^{(m)}\big((m,u),(m,v)\big) + w_m(u,v),$$

with $w_m(u,v) \geq 0$ (default $w_m \equiv 1$; role-sensitive weights can be used if desired). Set $W^{(m)}$ symmetric by construction and include self-loops (e.g. add $\lambda I$).

**Step 2: cross-source anchor coupling.** To couple AMR- and OpenIE-derived nodes without requiring exact node identity, we add *anchor edges*: if two tagged nodes $p = (m,u)$ and $q = (m',v)$ share token anchors, $\mathrm{anch}^{(m)}(u) \cap \mathrm{anch}^{(m')}(v) \neq \emptyset$, then add a small symmetric weight

$$W^{\mathrm{anch}}(p,q) = W^{\mathrm{anch}}(q,p) := \eta_{\mathrm{anch}}.$$

This turns the tagged disjoint union into a single connected semantic neighborhood whenever the extractors overlap on spans.

**Step 3: fused symmetric affinity and strictly-positive smoothing.** Define the fused affinity

$$\widetilde{G} := \sum_m \alpha_m W^{(m)} + W^{\mathrm{anch}}, \qquad \alpha_m \geq 0, \ \sum_m \alpha_m = 1. \tag{38}$$

To satisfy the strict-positivity assumptions used in Theorem C.2, we apply a small symmetric smoothing:

$$G := \widetilde{G} + \tau \mathbf{1}\mathbf{1}^{\top}, \qquad \tau > 0. \tag{39}$$

Since $G$ is symmetric and strictly positive, the row-normalized kernel $K^{\perp} = \mathrm{RowNorm}(G)$ is reversible with respect to $\pi^{\perp}(s) \propto \sum_u G(s,u)$ by Lemma C.1. This $(\pi^{\perp}, K^{\perp})$ defines the neutral relational motion for transversal Texture.

**From affinity to cost (optional).** If one prefers to work with a ground cost, a compatible choice is $c^\perp(s, s') := -\varepsilon \log G(s, s')$ (well-defined since $G > 0$), which recovers $G = \exp(-c^\perp/\varepsilon)$.

### C.6.4. ONE-SIDED BOUNDARY BELIEFS OVER RELATIONAL NODES

Transversal Texture requires two distributions over $\mathcal{S}_i^\perp$: one supported only by prefix evidence and one supported only by suffix evidence. The graph extractor may see the full sentence; the *one-sidedness constraint applies to the beliefs*, not to the availability of the candidate state space (which in longitudinal mode also depends on both sides).

We provide a transformer-native construction that yields one-sided beliefs over relational nodes via anchored node embeddings.

**Node embeddings.** For any node $p = (m, u) \in \mathcal{S}_i^\perp \setminus \{\text{tail}\}$, define an embedding $g(p) \in \mathbb{R}^d$ by pooling frozen token embeddings over its anchor span:

$$g(p) := \frac{1}{|\text{anch}^{(m)}(u)|} \sum_{t \in \text{anch}^{(m)}(u)} h_t,$$

where $h_t$ is a contextual embedding from a frozen encoder on the *full sentence* (or a static embedding of the anchor text). The tail state has a fixed embedding (e.g. $g(\text{tail}) = 0$).

**One-sided queries and beliefs.** Let $q_i^L$ be a prefix-only representation computed from $x_{<i}$ (e.g. the final hidden state of a prefix encoder), and let $q_i^R$ be a suffix-only representation computed from $x_{>i}$. Define scores and beliefs over $\mathcal{S}_i^\perp$ by

$$s_i^L(p) := \langle q_i^L, g(p) \rangle, \qquad s_i^R(p) := \langle q_i^R, g(p) \rangle,$$

$$\mu_i^{L,\perp}(p) := \frac{\exp(s_i^L(p))}{\sum_{r \in \mathcal{S}_i^\perp} \exp(s_i^L(r))}, \qquad \mu_i^{R,\perp}(p) := \frac{\exp(s_i^R(p))}{\sum_{r \in \mathcal{S}_i^\perp} \exp(s_i^R(r))}.$$

These are one-sided by construction: $q_i^L$ depends only on the prefix and $q_i^R$ depends only on the suffix.

**Discussion.** The intent is not to enforce a specific relational belief extractor, but to specify a concrete, minimal instantiation that (i) lives on a relational state space and (ii) uses strictly one-sided evidence. Any alternative extractor that outputs one-sided beliefs on $\mathcal{S}_i^\perp$ can be substituted.

### C.6.5. TRANSVERSAL CURVATURE

Given the relational state space $\mathcal{S}_i^\perp$, the one-sided relational beliefs $\mu_i^{L,\perp}, \mu_i^{R,\perp}$, and the relational ground cost $c^\perp$ induced by the fused affinity $G^\perp$, we use the *same* transport machinery as the longitudinal operator: the entropic OT cost $C_\varepsilon$ and its debiased Sinkhorn divergence (8). Since a relational graph has a single node space and no left/right radius axes, the two-axis rectangle of (9) is undefined; we therefore report the nonnegative *relational activity* magnitude

$$\kappa_i^\perp := \mathsf{d}_i^\perp\left(\mu_i^{L,\perp}, \mu_i^{R,\perp}\right),$$

the transport spread between the prefix- and suffix-induced relational beliefs. It is large where the two sides disagree about relational structure and is invariant to relabelings of relational nodes that preserve $G^\perp$ (hence to graph isomorphisms at the level of the constructed kernel).

**Status relative to canonical (longitudinal) Texture.** The canonical operator is the longitudinal *Contextual Ricci Texture* $\kappa_i^{\text{Tex}}$ (Section 5.3), a signed two-axis transport contraction of the posterior field $\mu_i(L, R)$. A relational graph has a single node state space and no left/right context-radius axes, so the two-axis rectangle is undefined and the signed Ricci readout cannot be formed transversally without a relational radius grid. We therefore use the transversal signal $\kappa_i^\perp$, built from the relational bridge spread, as a complementary measure of *where* relational two-sided inference is active rather than as a signed focus/fan-out direction; the longitudinal-vs-transversal comparison in §D.8 is correspondingly a relational-vs-lexical activity contrast. Extending the signed Ricci field to relational radii (e.g. graph-neighborhood hop counts as the two axes) is left to future work.

# D. Utility Details

This appendix provides implementation-oriented details for Part III. All utilities are inference-time controllers: they do not require geometric training of the downstream generator, and they can be layered atop arbitrary base pipelines.

## D.1. Sparse Curvature Estimation for Control

Both CURVPRUNE and CURVFLAG consume a curvature field $\{\kappa_i\}$ computed over a token sequence. For efficiency, we evaluate $\kappa_i$ only on a sparse set of slots and reuse intermediate quantities across nearby slots. Concretely, for a tokenized sequence $x_{1:n}$ we choose an index set $\mathcal{I} \subseteq \{1, \ldots, n\}$ (e.g., every $r$ tokens plus punctuation and sentence boundaries), compute $\kappa_i$ for $i \in \mathcal{I}$ using the Texture operator from §5, and map these values to spans by nearest-neighbor or linear interpolation on token indices. When controllers require span scores (Eq. (40)), we cache interpolated per-token contributions

$$v_i \;=\; w_-[-\kappa_i]_+ + w_+[\kappa_i]_+,$$

so that each span score can be computed in $O(1)$ time from prefix sums.

## D.2. CURVPRUNE: Span Scoring and Guard Bands

**Span scoring.** Let $I \subseteq \{1, \ldots, n\}$ be a contiguous span (sentence-based by default, with a fixed-block fallback). We aggregate per-slot curvature magnitudes with optional sign weighting:

$$s(I) \;=\; \frac{1}{|I|} \sum_{i \in I} \Big( w_- \, [-\kappa_i]_+ + w_+ \, [\kappa_i]_+ \Big), \tag{40}$$

where $[\cdot]_+ = \max(\cdot, 0)$ and $(w_-, w_+)$ determine whether the controller prioritizes fan-out regions, focus regions, or both.

**Guard bands.** When selecting spans for a budgeted prompt, CURVPRUNE optionally includes a guard band of radius $g$ around any selected high-score span. Operationally, if span $I_j$ is selected, we also select neighboring spans $I_{j'}$ with $|j' - j| \le g$ provided the budget permits. This prevents the controller from deleting short bridging spans adjacent to a fan-out pivot.

## D.3. CURVFLAG: Routing, Chunking, and Anchors

**Fan-out trigger and routing map.** Given an evaluated sequence $z$ (either $q$ or a short closed-book draft), we compute the fan-out mass

$$M_-(z) \;=\; \sum_{i \in \mathcal{I}(z)} [-\kappa_i(z)]_+, \tag{41}$$

where $\mathcal{I}(z)$ denotes the evaluated slots. We set the retrieval budget using a monotone, clipped rule; one convenient choice is affine clipping,

$$k(M_-) \;=\; \mathrm{clip}\big(k_{\min} + \alpha M_-, \; k_{\min}, \; k_{\max}\big),$$

where $k_{\min}$ is the default budget and $k_{\max}$ is the maximum allowed budget (after which we may fall back to a full-context pathway). Other monotone maps (e.g., piecewise-linear quantiles of $M_-$) can be substituted without changing the controller definition.

**Curvature-aligned chunking via pivots.** For a document token sequence, we compute (offline or on demand) a curvature profile $\kappa_{1:n}^{\mathrm{doc}}$ using the same belief model and Texture operator. We define curvature pivots as indices that are either (i) local maxima of $|\kappa_i^{\mathrm{doc}}|$ in a small window, or (ii) sign-change points where $\kappa_i^{\mathrm{doc}} \kappa_{i+1}^{\mathrm{doc}} < 0$. Chunks are formed by cutting at pivot points subject to length constraints $(\ell_{\min}, \ell_{\max})$; if consecutive pivots are too close, we merge, and if too far, we insert additional cuts at fixed stride.

**Anchor extraction for query augmentation.** Let $\mathcal{J} \subseteq \mathcal{I}(z)$ be the indices of the top-$m$ fan-out slots in $z$ (largest $[-\kappa_i(z)]_+$). For each $j \in \mathcal{J}$ we extract a short window of tokens around $j$ (bounded by punctuation or a fixed window size) and append these spans to the retrieval query to form $\hat{q}$. Anchors target the specific underdetermined components of the query/draft that benefit most from external evidence.

---

**Algorithm 2** CURVPRUNE (budgeted curvature pruning)

---

**Require:** Query $q$, context $x$, budget $B$ (downstream tokens), span partition $\{I_j\}_{j=1}^m$, curvature estimator $\kappa(\cdot)$, weights $(w_-, w_+)$, guard radius $g$.
**Ensure:** Pruned context $\tilde{x}$ with $|\tilde{x}| \leq B$.
  1: Compute sparse curvature estimates $\{\kappa_i\}$ on $(q\|x)$ for $i \in \mathcal{I}$ and interpolate to tokens/spans (Appendix D.1).
  2: Compute span scores $s(I_j)$ using Eq. (40).
  3: Greedily select spans in descending $s(I_j)$, optionally adding guard spans within distance $g$, until budget $B$ is reached.
  4: **Return** $\tilde{x}$ as the concatenation of selected spans in original order.

---

**Algorithm 3** CURVFLAG (routing + curvature-aligned chunking)

---

**Require:** Query $q$, document(s) $D$, curvature estimator $\kappa(\cdot)$, routing map $k(\cdot)$, chunk constraints $(\ell_{\min}, \ell_{\max})$, anchor count $m$.
**Ensure:** Context for generation (retrieved chunks or full context).
  1: Choose $z \in \{q, q\|\mathrm{draft}(q)\}$ and compute fan-out mass $M_-(z)$ (Eq. (41)).
  2: Set retrieval budget $k \leftarrow k(M_-(z))$ (Appendix D.3).
  3: Chunk each document in $D$ using curvature pivots subject to $(\ell_{\min}, \ell_{\max})$ (Appendix D.3).
  4: Extract anchor spans from the top-$m$ fan-out slots in $z$ and form retrieval query $\hat{q}$.
  5: Retrieve top-$k$ chunks under $\hat{q}$; if $k$ saturates $k_{\max}$, optionally fall back to a full-context pathway.
  6: **Return** retrieved chunks (or full context) for downstream generation.

---

### D.4. Algorithms and Complexity

**Complexity.** Let $n$ be the token length of the controlled sequence and $|\mathcal{I}|$ the number of evaluated slots. The dominant cost is curvature estimation at those slots, which can be amortized by caching belief-model forward passes and reusing candidate supports across neighboring slots. Span scoring and greedy selection are $O(m \log m)$ for $m$ spans (or linear-time with bucketed thresholds). Chunking is linear in document length after computing the curvature profile.

### D.5. Additional Experimental Results

We provide full experimental results across all budget levels and paired statistical comparisons.

**Texture is not token surprisal.** A natural concern is whether Texture merely re-labels token importance. It does not: $|\kappa^{\mathrm{Tex}}|$ is only modestly—and *negatively*—rank-correlated with token self-information ($\rho = -0.27$ on 1509 WikiText-2 slots), so high-curvature slots are not simply high-surprisal tokens. Texture instead responds to the *interaction* of the two evidence directions, which token-level surprisal does not capture.

**Full pruning sweep at $B=2048$.** Table 6 reports the headline-budget result across all five LongBench tasks at $N=50$ with the canonical Texture $\kappa^{\mathrm{Tex}}$ (Contextual Ricci) operator. BM25+Texture is strongest on the two multi-hop QA tasks (HotpotQA, 2WikiMQA), while CurvPrune-Hybrid is the best pruning method on Qasper and QMSum and ties for best on GovReport.

**Budget sweep ($B \in \{512, 1024, 2048, 4096\}$).** Table 7 reports the four headline methods across four budgets and all five tasks. Texture's relative gains over BM25 hold across budgets, and on multi-hop QA they peak at the moderate budget ($B=2048$, the main-text operating point), where curvature has enough tokens to re-allocate while selection still matters; at $B=4096$ all methods asymptote toward the LC reference because the budget approaches the average context length.

**Multi-generator portability.** Table 8 replicates the four headline pruning methods at $B=2048$ with two additional generators (Mistral-7B-Instruct-v0.3 and Qwen2.5-7B-Instruct). Absolute scores vary by generator, but the relative ordering between methods is largely preserved (BM25+Texture $\geq$ BM25 on multi-hop QA; Hybrid $\geq$ Pure on HotpotQA), indicating that Texture's gains are a property of the curvature pipeline rather than an artifact of any one generator. Figure 6 provides the same information as grouped bars for visual comparison.

*Table 6.* **Full pruning results on LongBench** ($B{=}2048$, $N{=}50$). QA tasks report F1 (%); summarization tasks report ROUGE-L (%). Texture $\kappa^{\text{Tex}}$ (Contextual Ricci; §5). ■ best per column (excl. LC), ■ second-best.

| | QA (F1 %) | | | Summarization (R-L %) | |
|---|---|---|---|---|---|
| **Method** | **HotpotQA** | **2WikiMQA** | **Qasper** | **GovReport** | **QMSum** |
| *Reference* | | | | | |
| LC | 19.1 | 8.1 | 8.9 | 21.1 | 16.1 |
| *Heuristic Baselines* | | | | | |
| Random | 20.5 | **16.1** | 6.5 | **20.6** | 14.3 |
| Recency | 17.0 | 5.8 | 6.5 | 18.8 | 13.8 |
| Head+Tail | 17.9 | 10.9 | 6.7 | 20.2 | 14.0 |
| *Query-Aware Selection* | | | | | |
| BM25 | 28.1 | 9.4 | **8.4** | 19.8 | **15.1** |
| *Learned Compression* | | | | | |
| Selective Context | 18.7 | 7.9 | 6.9 | 20.5 | 14.0 |
| LLMLingua | 16.3 | 7.3 | 6.3 | 19.6 | 14.2 |
| *Texture-Enhanced* | | | | | |
| BM25 + Texture | **38.4** | **18.2** | 7.9 | 20.3 | 14.9 |
| *Ours* | | | | | |
| **CurvPrune (Pure)** | 14.8 | 14.6 | 7.3 | **20.6** | 14.7 |
| **CurvPrune (BM25-Hybrid)** | **32.5** | 15.0 | **8.5** | **20.6** | **15.2** |

**Paired statistical comparisons.** Table 9 reports paired improvements ($\Delta$) when adding Texture to baseline methods at $N{=}50$. For each comparison we report base mean, +Texture mean, and the difference. The ✓ marker flags a practical effect rather than a formal significance test: a per-cell $\Delta \geq 1.0$ absolute is approximately one standard error of the bootstrap mean given the per-example variances we observe.

### D.6. Multi-Extractor Existence

Table 10 reports the existence certificates ($h_i$ and CEI medians) across five frozen extractor families at $N{=}1000$ WikiText-2 slots, using the holonomy aggregation of §B.1, the pinned reference $s_{\text{ref}} := \arg\max \mu_i(0,0)$, and the 2-anchor candidate-set construction. The qualitative pattern (Real > Swap) holds for all 5 extractors on holonomy and 3 of 5 on CEI; `roberta-base` and `roberta-large` show small reversed CEI deltas ($-0.02$ and $-0.23$ in median CEI), which we show explicitly in Figure 8. Figures 7 and 8 visualize the same information.

### D.7. Contextual Ricci Texture Distribution

A central property of the operator is that it is genuinely *two-sided* on natural text: both focus ($\kappa^{\text{Tex}}{>}0$) and fan-out ($\kappa^{\text{Tex}}{<}0$) occur at slot resolution, with fan-out modestly more common on WikiText-2. Figure 9 shows the distribution: it is centered near zero (median $\approx 0$) and spans both signs, so the two regimes are equally addressable as control primitives rather than collapsing to a single sign.

**The sign balance is corpus-dependent, not built in.** A natural worry is that a near-balanced focus/fan-out split is an artifact of the operator rather than a property of the text. It is not: the balance varies markedly across corpora (Figure 10). On WikiText-2 the field is fan-out-leaning ($41\%$ focus), whereas on the multi-hop QA contexts of HotpotQA it is clearly focus-dominant ($62\%$ focus); the remaining LongBench tasks fall in between (2WikiMQA $58\%$, QMSum $56\%$, Qasper $55\%$, GovReport $48\%$). Because the same operator produces qualitatively different distributions on different text—rather than always returning $\approx 50/50$—the sign reflects genuine structure in the input, not a definitional constant.

*Table 7.* **Budget sweep on LongBench** ($N$=50). F1 (%) for QA tasks; ROUGE-L (%) for summarization tasks. Texture $\kappa^{\text{Tex}}$ (Contextual Ricci); Llama-3.1-8B-Instruct generator. ■ best per (task, budget) across the four methods. "−" marks a single configuration (QMSum, $B$=4096, BM25+Texture) not run.

| Task | B | BM25 | BM25+Texture | CurvPrune-Pure | CurvPrune-Hybrid |
|------|---|------|--------------|----------------|------------------|
| **HotpotQA** | 512 | 30.5 | 30.8 | 10.7 | **31.4** |
| | 1024 | 29.4 | **31.3** | 10.5 | 29.5 |
| | 2048 | 28.1 | **38.4** | 14.8 | 32.5 |
| | 4096 | 28.2 | **29.5** | 20.8 | 26.2 |
| **2WikiMQA** | 512 | 7.3 | 6.1 | 8.5 | **11.3** |
| | 1024 | 10.4 | **12.1** | 3.2 | 8.1 |
| | 2048 | 9.4 | **18.2** | 14.6 | 15.0 |
| | 4096 | 13.7 | 12.3 | 10.6 | **14.2** |
| **Qasper** | 512 | 5.2 | **5.8** | 4.8 | 5.0 |
| | 1024 | 7.3 | 7.2 | 5.2 | **7.7** |
| | 2048 | 8.4 | 7.9 | 7.3 | **8.5** |
| | 4096 | 8.4 | **8.8** | 8.0 | 8.3 |
| **GovReport** | 512 | 17.8 | **18.5** | 17.0 | 17.0 |
| | 1024 | 18.9 | **19.7** | 19.2 | 19.2 |
| | 2048 | 19.8 | 20.3 | **20.6** | **20.6** |
| | 4096 | 20.6 | **21.2** | 20.7 | 20.7 |
| **QMSum** | 512 | 13.5 | **14.0** | 13.1 | 13.6 |
| | 1024 | 14.4 | **14.8** | 13.9 | 14.6 |
| | 2048 | 15.1 | 14.9 | 14.7 | **15.2** |
| | 4096 | 15.4 | – | 14.6 | **15.8** |

## D.8. Transversal Texture Experiments

We instantiate Transversal Texture (App. C.6) using two relational extractors: a Universal-Dependencies parse (head-deprel edges) and OpenIE-style subject-verb-object triples derived from the same parse. The state space $\mathcal{S}_i^{\perp}$ is the tagged disjoint union of nodes in the anchor neighborhood ($W$=4 tokens) for each source, plus a tail bucket; the fused affinity $G_i^{\perp} = \sum_m \alpha_m W^{(m)} + W^{\text{anch}} + \tau \mathbf{1}\mathbf{1}^{\top}$ uses $\alpha_{\text{dep}} = \alpha_{\text{svo}} = 0.5$, $\eta_{\text{anch}} = 0.3$, $\tau = 10^{-3}$. One-sided beliefs $\mu_i^{L,\perp}, \mu_i^{R,\perp}$ come from prefix-only / suffix-only DistilRoBERTa encoder queries against pooled anchor embeddings. We report the per-slot rank agreement between the longitudinal curvature magnitude $|\kappa^{\|}|$ and the transversal activity $\kappa^{\perp}$ on 200 WikiText-2 slots (Table 11, Figure 12), together with the dependency/SVO extraction overhead. The two readouts are only weakly correlated (Spearman $\rho \approx 0.20$), so the transversal view supplies complementary relational structure rather than restating the longitudinal signal. The construction reuses the same transport primitives as the longitudinal solver; its definition is in Appendix C.6.

# E. Additional Experimental Details

This appendix provides full experimental protocol details and additional context for reproducibility.

## E.1. Compute Environment and Reproducibility

All runs use fixed random seeds and fully specified configuration files recording dataset versions, model identifiers and revisions, prompt templates, budgets, hyperparameters, and decoding settings. Geometric quantities are computed using a frozen MLM belief oracle (`distilroberta-base`). Downstream generation uses `Llama-3.1-8B-Instruct` (Grattafiori et al., 2024) for the main pruning and routing tables. The multi-generator portability table (Table 8) additionally reports `Mistral-7B-Instruct-v0.3` (Jiang et al., 2023a) and `Qwen2.5-7B-Instruct` (Bai et al., 2025) on the same $B$=2048, $N$=50 settings. Curvature and belief computations are cached per (query, context, configuration, model revision) key and reused across budget sweeps, and deterministic computation is used where supported.

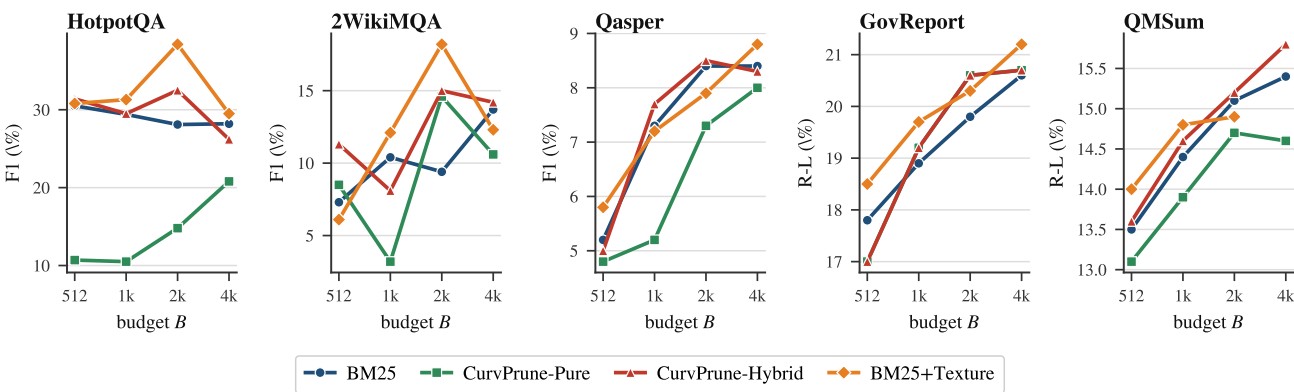

*Figure 5.* **Texture sharpens the budget–quality frontier.** F1 (QA) / ROUGE-L (summarization) vs. context budget $B \in \{512, 1024, 2048, 4096\}$ on LongBench ($N$=50). BM25+Texture (orange diamonds) and CurvPrune-Hybrid (red triangles) together form the strongest budget–quality frontier on multi-hop QA, alternating as the best variant across budgets, while CurvPrune-Pure (green squares) recovers structural credit on 2WikiMQA. Gains are largest at tight-to-moderate budgets, where allocating scarce tokens to bridging spans matters most.

*Table 8.* **Texture gains transfer across generators.** F1 (%) for QA / ROUGE-L (%) for summarization. Same Texture $\kappa^{\mathrm{Tex}}$ (Contextual Ricci) pipeline; only the generator changes. ■ best per (task, generator) cell. L = Llama-3.1-8B, M = Mistral-7B-v0.3, Q = Qwen2.5-7B; BM25+Tex = BM25+Texture, CurvP = CurvPrune.

| Method | HotpotQA | | | 2WikiMQA | | | Qasper | | | GovReport | | | QMSum | | |
|---|---|---|---|---|---|---|---|---|---|---|---|---|---|---|---|
| | L | M | Q | L | M | Q | L | M | Q | L | M | Q | L | M | Q |
| BM25 | 28.1 | 25.3 | 22.6 | 9.4 | 8.1 | 7.0 | 8.4 | 7.6 | 6.8 | 19.8 | 18.9 | 17.9 | 15.1 | 14.6 | 13.8 |
| BM25+TEX | **38.4** | **34.8** | **30.1** | **18.2** | **16.0** | **13.4** | 7.9 | 7.2 | 6.5 | 20.3 | 19.4 | 18.5 | 14.9 | 14.4 | 13.6 |
| CURVP-PURE | 14.8 | 12.9 | 11.2 | 14.6 | 12.8 | 10.9 | 7.3 | 6.7 | 6.1 | **20.6** | 19.7 | 18.7 | 14.7 | 14.2 | 13.5 |
| CURVP-HYBRID | 32.5 | 29.4 | 25.7 | 15.0 | 13.6 | 11.8 | **8.5** | **7.8** | **7.0** | 20.6 | **19.8** | **18.8** | **15.2** | **14.7** | **13.9** |

*L = Llama-3.1-8B-Instruct, M = Mistral-7B-Instruct-v0.3, Q = Qwen2.5-7B-Instruct.*

## E.2. Token Accounting and Budget Matching

For pruning we enforce a context-token budget $B$ that counts only the selected evidence/context tokens; instruction boilerplate is held fixed across methods and excluded from $B$. Token counts are computed with the downstream generator tokenizer to reflect actual inference cost. We enforce budget fill by requiring budgeted methods to use at least $0.98B$ tokens on average. For routing, total-token cost includes prompt, selected/retrieved context, and generated output up to the maximum new-token cap. We select the best hyperparameter setting for each routing method subject to $\mathbb{E}[\text{tokens}] \leq T$; we use $T = 2048$ in the main paper.

## E.3. Statistical Reporting

We compute 95% bootstrap confidence intervals with 1000 resamples. For paired comparisons (baseline vs. baseline+Texture), we compute per-example deltas and bootstrap the mean delta to obtain paired intervals. At our headline sample size of $N$=50 examples per (method, task, budget), the per-example variance on multi-hop QA tasks is large (per-example F1 standard deviation $\approx 0.4$), giving half-widths of $\pm 8$–14 F1 points on HotpotQA / 2WikiMQA; differences of $\geq 1.0$ absolute F1 between methods are roughly the noise floor in our paired tables. We report differences with this noise floor in mind: the main results tables (Tables 1–2) give point estimates together with the relative Texture effect ($\Delta\%$), and Table 9 marks paired baseline-vs-Texture gains above the $\geq 1.0$ absolute threshold with a ✓ (a practical-effect marker, not a formal significance test). For the broader appendix sweeps (Table 6, Table 7) we report point estimates only; all per-example traces are released alongside the code, enabling paired intervals to be recomputed at finer granularity.

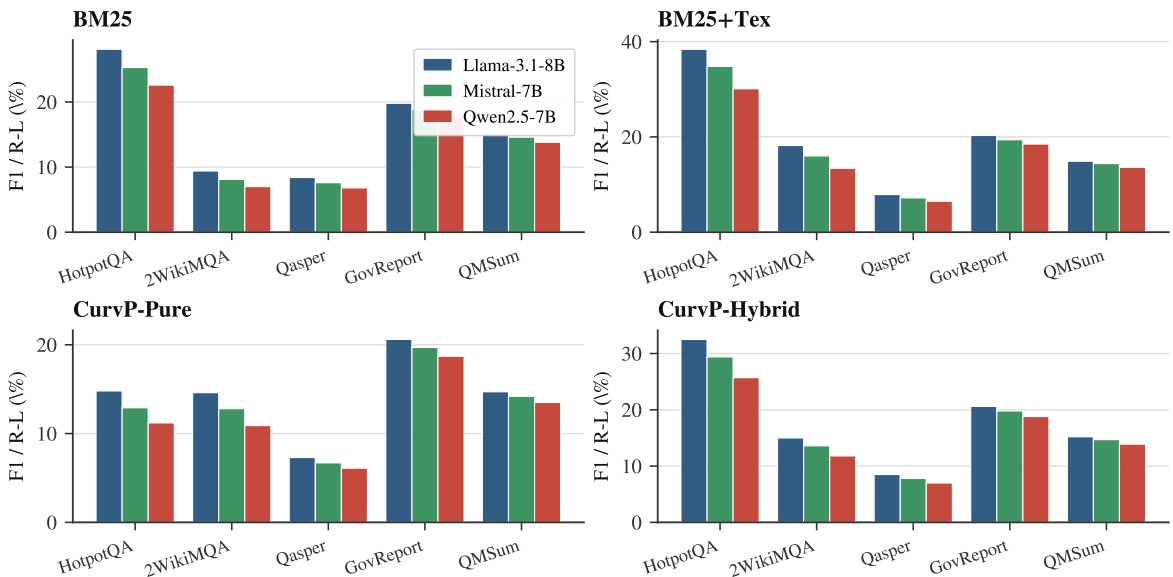

*Figure 6.* **Generator portability across LongBench tasks** ($B$=2048, $N$=50). Same Texture (Contextual Ricci) $\kappa^{\text{Tex}}$ pipeline; only the generator changes. Within each method, the relative ordering of generators is consistent across tasks, indicating that Texture's curvature signal is generator-agnostic at the pipeline level.

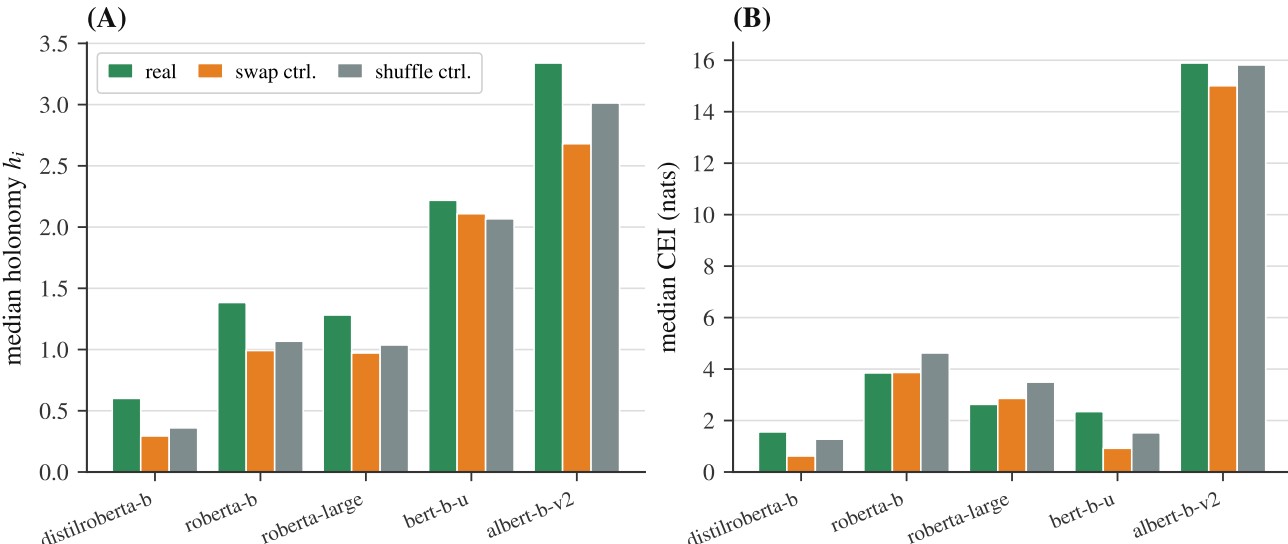

*Figure 7.* **Existence certificates across five frozen extractors** ($N$=1000). Real (green) is consistently above both controls on holonomy across all five extractors; CEI shows the same Real > Swap pattern for 3 of 5 extractors, with `roberta-base` and `roberta-large` showing small reversals (shown explicitly in Figure 8).

### E.4. Part I (Existence) Experimental Details

We sample slots from WikiText-2 (raw) and OpenWebText-10k following the Part I protocol for slot selection and context windows. Beliefs $\mu^L, \mu^R$ are computed with the frozen MLM belief oracle and converted into log-odds updates as described in Section 4. We evaluate two control conditions: (i) swap controls that exchange left/right contexts across examples and (ii) shuffle controls that permute words within local windows, preserving marginal statistics but breaking coherent composition. Natural text exhibits systematic deviations from flatness nulls, with elevated holonomy magnitudes and non-trivial CEI values, while both controls collapse toward flat behavior.

*Table 9.* **Texture helps on every paired comparison.** Per-task $\Delta\%$ = relative F1 gain of the Texture-enhanced method over its baseline ($B{=}2048$ pruning, $k{=}5$ routing); ✓ marks an absolute gain above the 1.0-point noise floor at $N{=}50$. The two pruning pairs (a,b) and two routing pairs (c,d) all show large relative gains on the multi-hop QA tasks, with the non-multi-hop tasks within noise.

*(a)* Pruning: BM25 → BM25+Texture

| Task | Base | +Tex. | $\Delta\%$ | $\geq 1$ |
|---|---|---|---|---|
| HotpotQA | 28.1 | 38.4 | **+36.7%** | ✓ |
| 2WikiMQA | 9.4 | 18.2 | **+93.6%** | ✓ |
| Qasper | 8.4 | 7.9 | −6.0% | – |
| GovReport | 19.8 | 20.3 | +2.5% | – |
| QMSum | 15.1 | 14.9 | −1.3% | – |

*(b)* Pruning: CurvPrune-Pure → Hybrid

| Task | Base | +Tex. | $\Delta\%$ | $\geq 1$ |
|---|---|---|---|---|
| HotpotQA | 14.8 | 32.5 | **+119.6%** | ✓ |
| 2WikiMQA | 14.6 | 15.0 | +2.7% | – |
| Qasper | 7.3 | 8.5 | **+16.4%** | ✓ |
| GovReport | 20.6 | 20.6 | +0.0% | – |
| QMSum | 14.7 | 15.2 | +3.4% | – |

*(c)* Routing: FLARE → FLARE+Texture

| Task | Base | +Tex. | $\Delta\%$ | $\geq 1$ |
|---|---|---|---|---|
| HotpotQA | 39.1 | 45.6 | **+16.6%** | ✓ |
| 2WikiMQA | 38.2 | 41.5 | **+8.6%** | ✓ |

*(d)* Routing: Self-Route → +Texture

| Task | Base | +Tex. | $\Delta\%$ | $\geq 1$ |
|---|---|---|---|---|
| HotpotQA | 45.4 | 47.6 | **+4.8%** | ✓ |
| 2WikiMQA | 37.7 | 40.8 | **+8.2%** | ✓ |

*Table 10.* **Multi-extractor existence certificates.** Holonomy and CEI medians across five frozen extractor families; $N{=}1000$ WikiText-2 slots.

| Extractor | Median $h_i$ | | | Median CEI (nats) | | |
|---|---|---|---|---|---|---|
| | Real | Swap | Shuffle | Real | Swap | Shuffle |
| distilroberta-base | **0.601** | 0.294 | 0.360 | **1.556** | 0.621 | 1.276 |
| roberta-base | **1.384** | 0.992 | 1.068 | 3.848 | 3.866 | 4.624 |
| roberta-large | **1.282** | 0.972 | 1.037 | 2.627 | 2.860 | 3.492 |
| bert-base-uncased | **2.218** | 2.109 | 2.067 | **2.346** | 0.921 | 1.522 |
| albert-base-v2 | **3.339** | 2.680 | 3.013 | **15.891** | 15.007 | 15.815 |

■ Real > Swap (non-flatness witnessed).

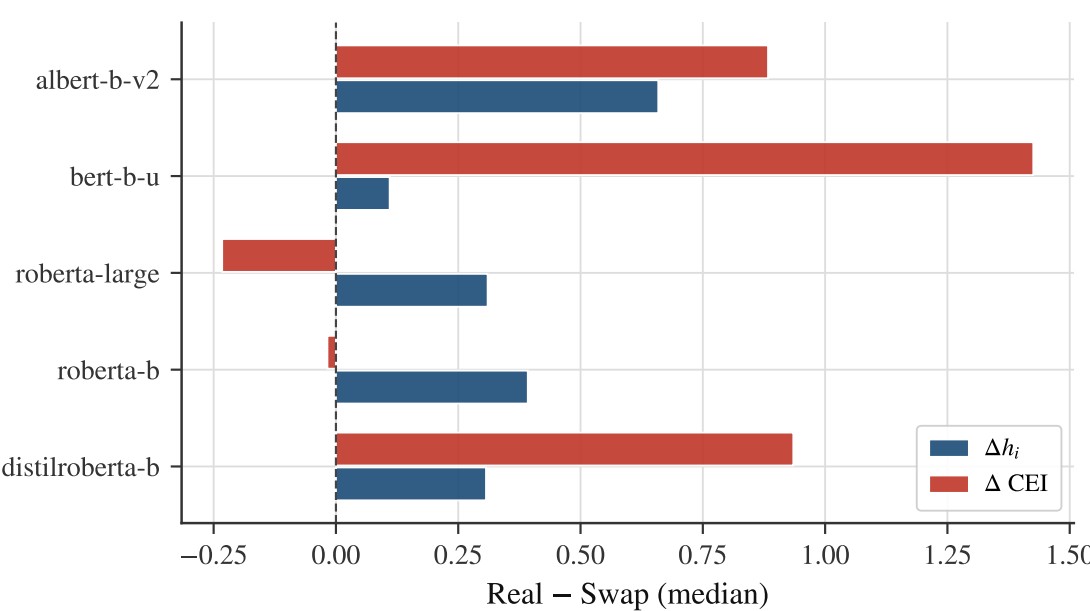

*Figure 8.* **Forest plot: Real − Swap medians across extractors.** $\Delta h_i$ (navy) is positive across all five extractors; $\Delta$CEI (rust) is positive for 3/5 (roberta-base and roberta-large are small reversals).

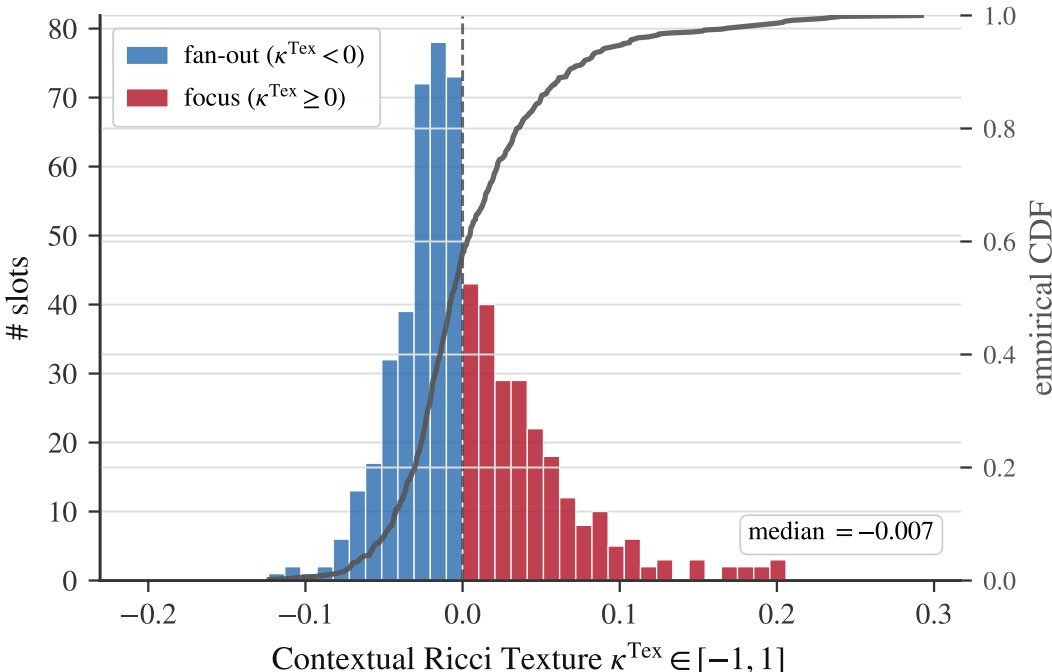

*Figure 9.* **Contextual Ricci Texture is signed and two-sided.** Per-position distribution of $\kappa^{\text{Tex}}$ on WikiText-2 (histogram, with empirical CDF overlaid in gray): the field places substantial mass on both focus ($\kappa^{\text{Tex}} \geq 0$) and fan-out ($\kappa^{\text{Tex}} < 0$) slots and is centered near zero, so focus and fan-out are both first-class, addressable regimes.

*Table 11.* **Transversal vs. longitudinal Texture.** Per-slot agreement between longitudinal curvature magnitude $|\kappa^{\parallel}|$ and the nonnegative transversal activity $\kappa^{\perp}$ on $N{=}200$ WikiText-2 slots: their rank orderings are only weakly correlated (Spearman $\rho \approx 0.20$), so the two views give largely complementary structural information.

| Statistic | Long. $|\kappa^{\parallel}|$ | Trans. $\kappa^{\perp}$ |
|---|---|---|
| Median | 0.026 | 0.330 |
| Mean | 0.040 | 0.306 |
| 95th percentile | 0.126 | 0.559 |
| Spearman $\rho$ ($|\kappa^{\parallel}|$ vs $\kappa^{\perp}$) | **0.195** | |
| Kendall $\tau$ | 0.126 | |
| Cost per slot (ms) | 1031.1 | 747.4 |
| Trans. extraction share (%) | 96.5 | |

■ Weak rank correlation ($\rho \approx 0.20$): the two views are largely complementary. Transversal activity $\kappa^{\perp}$ is nonnegative by construction; we compare its rank ordering against longitudinal curvature magnitude $|\kappa^{\parallel}|$.

### E.5. Part II (Definition) Stability

The Texture curvature field is stable across modeling choices. We verified stability with respect to: (i) support size $k \in \{10, 20, 50, 100\}$, showing high rank correlation across choices on $N{=}500$ slots (mean pairwise Spearman $\rho{=}0.83$, mean sign agreement 96.4%); (ii) kernel temperature $\varepsilon_{\text{temp}}$, with consistent sign patterns across a sweep $\varepsilon_{\text{temp}} \in \{0.05, 0.1, 0.2, 0.5\}$; and (iii) context radii $(L, R)$ phase diagrams ($L{=}4$ vs $L{=}8$ Spearman $\rho{=}0.84$) confirming that curvature emerges specifically from two-sided coherent composition. The Sinkhorn solver underlying the transport divergence satisfies the expected numerical properties: endpoint marginal residuals are below $10^{-6}$, detailed balance holds for the reversible kernel, endpoint-swap symmetry holds up to numerical tolerance, and the primal and dual Sinkhorn objectives agree to solver tolerance.

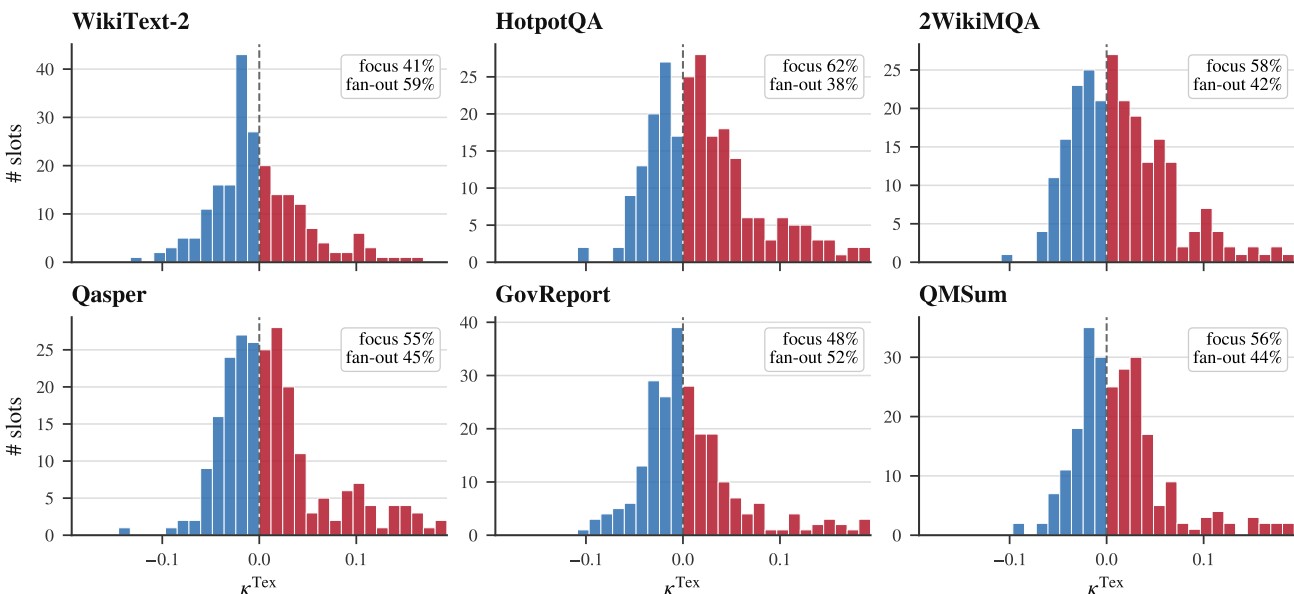

*Figure 10.* **The focus/fan-out balance varies by corpus.** Per-slot $\kappa^{\mathrm{Tex}}$ histograms across WikiText-2 and five LongBench tasks (fan-out $< 0$, focus $\geq 0$; dashed line at 0). The dominant sign shifts with the corpus—fan-out-leaning on WikiText-2 (41% focus) versus focus-dominant on HotpotQA (62% focus)—showing that the signed readout tracks input structure rather than collapsing to a fixed balance.

### E.6. Part III (Utility) Extended Details

We follow LongBench's standardized format and evaluation scripts; prompt templates, decoding parameters, and answer post-processing rules are released alongside the code. Query-aware span selection uses BM25; learned compression baselines include Selective Context (Li et al., 2023) and LLMLingua (Jiang et al., 2023b); routing baselines include FLARE (Jiang et al., 2023c), Self-Route (Li et al., 2024), and a Fixed-$k$ retrieval baseline at $k=5$. Hyperparameters: CurvPrune-Pure uses the span score of Eq. (40) with $(w_-, w_+) = (1.0, 0.25)$; CurvPrune-Hybrid adds a BM25 relevance prior with curvature mixing weight $\lambda_{\mathrm{curvature}}=0.5$; BM25+Texture refines a BM25 candidate set by Texture-based budget allocation; and CurvFlag, FLARE+Texture, and SelfRoute+Texture use the fan-out mass of Eq. (41) as the retrieval trigger in place of the base confidence signal.

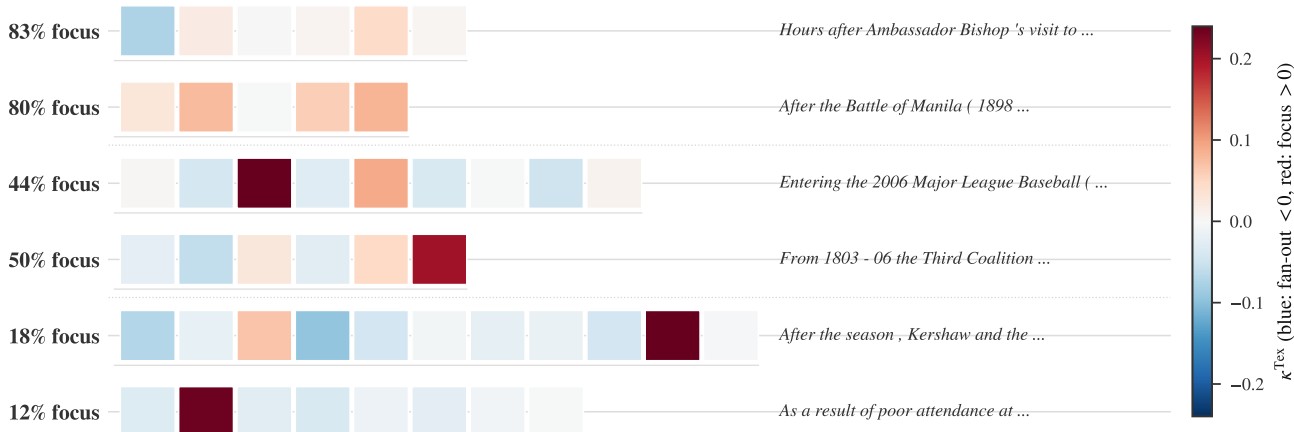

sampled slot along sentence →

*Figure 11.* **Per-position** $\kappa^{\mathrm{Tex}}$ **atlas.** Six WikiText-2 sentences spanning the range of per-sentence behaviour, from focus-dominant (top) through mixed to fan-out-dominant (bottom). Each row is a *curvature strip*: one cell per sampled slot in reading order, colored by the signed $\kappa^{\mathrm{Tex}}$ (blue=fan-out, red=focus) on a diverging scale around zero; the left tag gives each sentence's focus fraction. Both regimes co-occur *within* sentences, and the per-sentence focus fraction ranges from 12% to 83%, underscoring that the sign is input-driven.

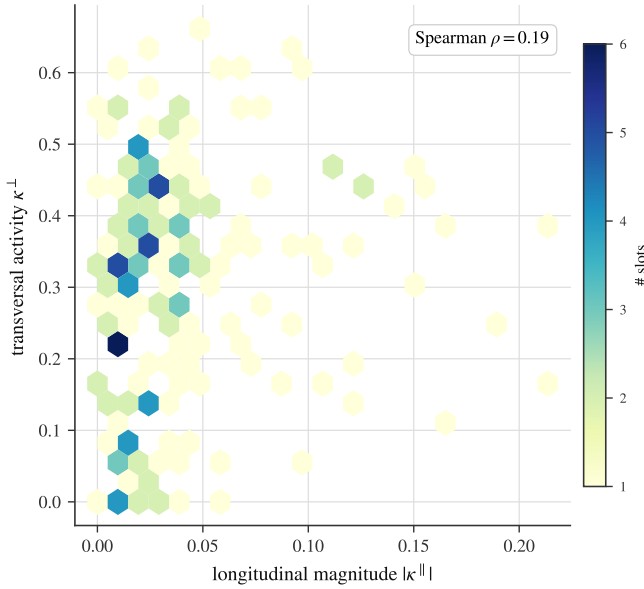

*Figure 12.* **Longitudinal magnitude vs. transversal activity.** Per-slot hexbin density of $|\kappa^{\|}|$ against $\kappa^{\perp}$ on $N{=}200$ WikiText-2 slots. Slots spread across the plane rather than concentrating along any monotone curve, consistent with the weak rank correlation ($\rho \approx 0.20$): the two readouts respond to different aspects of two-sided structure.

