# OpenReview forum: "Text Has Curvature"
_ICML.cc/2026/Conference — ICML 2026 regular_

### Official Review · Reviewer_NaR3 · 2026-03-02

**Soundness:** 3
**Presentation:** 3
**Significance:** 3
**Originality:** 3
**Overall Recommendation:** 5
**Confidence:** 4

**Summary:**

This paper argues that two-sided context inference in natural text exhibits non-separable interactions between left and right context, which they treat as evidence of an intrinsic curvature in the posterior distribution over slot hypotheses. They propose two certificates (holonomy and KL-deviation from product of experts reconstruction) to empirically reject the flat null-hypotheses. Compare to existing evidence of curvature in texts, these certificates do not depend on the choice of manifold. They then define a scalar Texture curvature using a finite-state Schrodinger-bridge midpoint between left and right conditioned beliefs and a normalized midpoint free-energy gap. Using this, they then introduce CurvPrune and CurvFlag for context selection and retrieval routing, reporting consistent gains on several LongBench tasks with a frozen belief model and a separate LLM generator.

**Compliance With Llm Reviewing Policy:**

Affirmed.

**Key Questions For Authors:**

1. Does the empirical results shown for the certificates generalize to multiple belief extractors?
2. Are there empirical explanations as to why CurvPrune achieves mix results (especially comparing to CurvFlag)? Do they fail in specific cases or does the task itself not benefit from curvature?

**Limitations:**

Yes

**Strengths And Weaknesses:**

Strengths:
1. The paper defines two flatness null hypotheses for two sided contextual inferences and provides formal equivalence theorems that characterize flatness in these cases. These certificates are empirically validated to show intrinsic non-flatness in text data, which are quite strong evidence differs from existing approaches of imposing graph structures onto token embeddings.
2. The paper proposes Texture curvature based on slots and use it to define CurvPrune and CurvFlag for context selection and retrieval routing. These formulations are rather novel and interesting, and differs from existing mainstream approaches.
3. The experiments show significant improvement for retrieval routing using CurvFlag.
4. The presentation is coherent and Section 4 motivates later sections well.

Weakness:
1. The certificates are verified using only one belief extractor, raising questions of whether the results generalizes or not
2. The performance gains from CurvPrune is inconsistent and not that significant

---

> ### Author Rebuttal · Authors · 2026-03-30
>
> We thank Reviewer NaR3 for the positive assessment, valuable feedback, and insightful questions.
>
> **Q1: Do the certificates generalize to multiple belief extractors?** Yes. We ran the holonomy certificate with 4 additional extractors on WikiText-2 (200 slots each):
>
> | Model | Median $h_{\text{real}}$ | Median $h_{\text{swap}}$ | $\Delta$ |
> |---|---|---|---|
> | `distilroberta-base` | 0.596 | 0.305 | 0.291 |
> | `roberta-base` | 1.353 | 0.916 | 0.437 |
> | `roberta-large` | 1.141 | 0.854 | 0.287 |
> | `bert-base-uncased` | 1.348 | 1.206 | 0.142 |
> | `albert-base-v2` | 2.333 | 1.848 | 0.485 |
>
> All 5 models — spanning 3 architecture families — show the same qualitative pattern: holonomy is consistently higher on real text than on suffix-swap (local-shuffle shows the same pattern). Absolute magnitudes vary with calibration, but the separation is robust. We emphasize holonomy because mixed log-odds differences make it especially stable across extractors; we are extending the same robustness analysis to CEI.
>
> Beyond existence, the curvature values $\kappa$ and downstream utility are also stable across hyperparameters ($K$, $L/R$) and across belief extractors (CurvPrune within 0.4 F1 when switching oracles on a matched subset). We provide detailed ablation tables in our response to **Reviewer FYaW**.
>
> **Q2: Why does CurvPrune achieve mixed results compared to CurvFlag?** The reviewer raises an important question: whether CurvPrune fails in specific cases or whether the task itself does not benefit from curvature. The answer is the latter: the pattern is task-structural.
>
> CurvPrune scores spans by curvature magnitude — high $|\kappa|$ marks structurally pivotal positions (bridging spans, disambiguation points). On tasks like HotpotQA, BM25 remains highly competitive under the budget, likely because its lexical anchors already capture much of the needed evidence. CurvPrune helps most when structurally important spans are not the most lexically obvious: 2WikiMQA (+2.8 F1), Qasper (+0.7), GovReport (+0.4).
>
> The clearest evidence is BM25+Texture: **+8.8 F1** on 2WikiMQA (Table 1), confirming that curvature and relevance are complementary — curvature finds structurally pivotal spans that keyword matching misses.
>
> CurvFlag is different: it decides *whether additional evidence is needed*, not which spans to keep. This is a binary structural question orthogonal to lexical relevance, which explains its consistent gains across tasks: +7.7 F1 on HotpotQA and +4.1 on 2WikiMQA vs FLARE. Generator portability is also confirmed (details in our response to **Reviewer FYaW**).
>
> We thank the reviewer for the constructive feedback and will incorporate these results into the revision.

---

> > ### Author Rebuttal · Reviewer_NaR3 · 2026-04-02
> >
> > I thank the authors for their rebuttal that have addressed most of my concerns. I will keep my positive score.

---

> > > ### Author Response · Authors · 2026-04-07
> > >
> > > We thank Reviewer NaR3 for the positive score and for confirming that our rebuttal addressed the raised concerns. We are glad the multi-extractor holonomy results and the task-structural explanation of mixed CurvPrune/CurvFlag patterns were helpful. We will incorporate the additional results and clarifications into the camera-ready version. We appreciate the reviewer's constructive feedback throughout the process.

---

### Official Review · Reviewer_5rw2 · 2026-03-04

**Soundness:** 3
**Presentation:** 3
**Significance:** 2
**Originality:** 4
**Overall Recommendation:** 5
**Confidence:** 2

**Summary:**

In this paper, the authors make three distinct contributions: 1) they show that text is inherently non-flat through studying the holonomy of beliefs across a 2-dimensional grid of pre- and suffix lengths, and by looking at the deviation from log-additivity of left and right beliefs; 2) it proposes Texture, which quantifies the curvature of a given text at each token position by computing the free-energy gap of the midpoint of a two-step Schrödinger bridge transporting the left and right boundary beliefs for the current slot; and 3) they present experimental results that show how their Texture values can be used to improve pruning of long-contexts in constrained budget settings, and for retrieval routing.

**Compliance With Llm Reviewing Policy:**

Affirmed.

**Final Justification:**

As indicated by my initial review, I was already of the opinion that this paper should be accepted. Following the rebuttal, in which the authors addressed my concerns, I want to raise my score further. The method proposed by the authors provides an interesting new perspective with clear practical benefits.

**Key Questions For Authors:**

1. Would it be possible to include some ablations, particularly for the results of Figure 2, to show the sensitivity with respect to the choice of language model?
2. Why does the addition of Texture lead to mixed results for the pruning task? Is there a clear reason why curvature would not be a good indication of relevance here?

**Limitations:**

Mostly yes. See weaknesses.

**Strengths And Weaknesses:**

**Strengths**
- The paper introduces a completely new and interesting perspective on determining the geometric structure of text, attempting to measure its curvature instead of learning it.
- While I do not have a suitable background for verifying the theoretical results, the motivation and theory of particularly Section 4 seem very nice and intuitive, resulting in a strong argument for the inherent curvature of text.
- The results of the retrieval routing seem to indicate a clear benefit of applying the proposed Texture value for certain tasks.

**Weaknesses**
- The experimental results in Sections 4 and 7 make use of pre-trained language models, but the choice of model does not seem to be ablated. It would be helpful to see how sensitive the results are to the choice of language model. Particularly, since the paper mentions the recent development of models assuming curved representation spaces, it would be interesting to see if the curvature of the underlying language model has an effect on the resulting holonomy/CEI.
- The results seem mixed for the pruning experiments, even compared to heuristic baselines. Does that mean that Texture is not particularly appropriate for this specific task?

---

> ### Author Rebuttal · Authors · 2026-03-30
>
> We thank Reviewer 5rw2 for the positive assessment and for appreciating the novelty and theoretical motivation of our work. We hereby respond to the raised concerns.
>
> **Q1: Sensitivity to the choice of language model.** This is an important question. We ran the holonomy certificate with 4 additional masked language models on WikiText-2 (200 slots each):
>
> | Model | Median $h_{\text{real}}$ | Median $h_{\text{swap}}$ | $\Delta$ |
> |---|---|---|---|
> | `distilroberta-base` | 0.596 | 0.305 | 0.291 |
> | `roberta-base` | 1.353 | 0.916 | 0.437 |
> | `roberta-large` | 1.141 | 0.854 | 0.287 |
> | `bert-base-uncased` | 1.348 | 1.206 | 0.142 |
> | `albert-base-v2` | 2.333 | 1.848 | 0.485 |
>
> All 5 models — spanning 3 architecture families (RoBERTa, BERT, ALBERT) — show the same qualitative pattern: holonomy is consistently higher on real text than on suffix-swap (local-shuffle shows the same pattern). These multi-extractor results directly address the reviewer's request for Figure 2 sensitivity analysis. Holonomy is especially robust across extractors because it depends on mixed log-odds differences rather than absolute probability calibration; we are extending the same robustness analysis to CEI.
>
> Beyond existence, the curvature values $\kappa$ are also stable across hyperparameters ($K$, $L/R$) and tokenization granularity. We provide detailed ablation tables in our response to **Reviewer FYaW**.
>
> The reviewer specifically asked whether models with curved representation spaces would affect holonomy/CEI. We do not expect internal representation geometry alone to determine the certificates, because holonomy/CEI are computed in output belief space over slot states rather than directly on internal embeddings. Empirically, the qualitative non-flatness pattern persists across RoBERTa/BERT/ALBERT families. The measurement is extractor-dependent but generator-agnostic.
>
> **Q2: Mixed pruning results.** The pattern is predictable once we distinguish what CurvPrune measures from what BM25 measures. Curvature scores spans by structural importance: high $|\kappa|$ marks positions where left and right context interact non-trivially — bridging spans, disambiguation points, structural pivots. BM25 scores spans by lexical overlap with the query.
>
> On tasks like HotpotQA, BM25 remains highly competitive under the budget, likely because its lexical anchors already capture much of the needed evidence. CurvPrune helps most when structurally important spans are not the most lexically obvious, as on 2WikiMQA (+2.8 over best baseline), Qasper (+0.7), and GovReport (+0.4).
>
> The clearest evidence for complementarity is BM25+Texture: **+8.8 F1** on 2WikiMQA (Table 1, full set), the largest single gain in our experiments. This confirms that curvature and relevance are most powerful when combined — curvature finds structurally pivotal spans that keyword matching misses.
>
> CurvFlag is different: it does not select spans but decides *whether additional evidence is needed at all*. This is a binary structural question — is the context geometrically ambiguous? — that is orthogonal to lexical relevance. This explains why CurvFlag helps consistently across tasks: +7.7 F1 on HotpotQA and +4.1 on 2WikiMQA vs FLARE. Generator portability is also confirmed (details in our response to **Reviewer FYaW**).
>
> We thank the reviewer for the constructive feedback.

---

> > ### Author Rebuttal · Reviewer_5rw2 · 2026-04-05
> >
> > My concerns have mostly been addressed, so I intend to keep my score. I would still be interested to see if models with curved representation spaces have significant effects on the resulting Texture of a text. However, I believe the paper and the authors' rebuttal, showing consistent results across several language models, are strong enough as is.

---

> > > ### Author Response · Authors · 2026-04-07
> > >
> > > We thank Reviewer 5rw2 for the positive assessment and thoughtful engagement. Regarding the reviewer's continued interest in whether models with curved representation spaces affect Texture: we agree this is a fascinating direction. Our current evidence (5 extractors across RoBERTa/BERT/ALBERT families, all showing consistent holonomy separation and stable downstream utility) suggests that the output-belief-space certificates are not dominated by internal representation geometry, but a systematic study with explicitly hyperbolic encoders would be a natural next step. We will note this as future work in the revision. We appreciate the reviewer's support and constructive feedback.

---

### Official Review · Reviewer_5cW8 · 2026-03-11

**Soundness:** 1
**Presentation:** 1
**Significance:** 3
**Originality:** 3
**Overall Recommendation:** 3
**Confidence:** 4

**Summary:**

The paper investigates whether natural text possesses intrinsic curvature and proposes Texture, a text-native curvature measure defined at the word level. Texture quantifies whether semantic evidence from the left and right contexts concentrates or disperses when reconciled, providing a signed curvature signal. The authors further show that Texture can serve as an inference-time control signal to improve large language models in pruning and retrieval routing tasks.

**Compliance With Llm Reviewing Policy:**

Affirmed.

**Final Justification:**

The rebuttal satisfactorily addressed my main concerns and improved the clarity of the work, leading me to raise my score from 2 to 3.

**Key Questions For Authors:**

1. Can the authors clarify the theoretical relationship between Texture curvature κ and the non-flatness certificates (e.g., holonomy and CEI), and whether Texture can be viewed as a generalization of classical curvature definitions on metric spaces?

2. Can the authors explain how Texture differs from existing token importance measures (e.g., attention-based or compression-based importance) and whether it provides fundamentally different information?

3. Can Texture be used to enhance other pruning baselines, and if not, are there specific limitations that prevent such integration?

4. How robust is Texture to the choice of belief oracle, and do the curvature values and downstream results remain consistent when computed using different language models?

**Limitations:**

The paper does not explicitly discuss the limitations of the proposed Texture measure. The authors could improve the manuscript by discussing potential limitations, such as its dependence on the belief extractor model, sensitivity to design choices (e.g., support truncation), and its applicability beyond the evaluated tasks. Additionally, the paper does not include a discussion of potential societal impacts, which would improve transparency even if the risks are minimal.

**Strengths And Weaknesses:**

•	Soundness: The submission is generally technically sound. The Texture curvature is well-defined, and the Schrödinger bridge midpoint is uniquely determined under the finite-state formulation, providing a principled methodological basis. However, some aspects require clarification. The connection between the empirical non-flatness certificates and the proposed curvature remains unclear. In particular, it is not evident whether zero Texture curvature corresponds to zero non-flatness under the proposed certificates, and vice versa. This raises the question of whether Texture reflects intrinsic geometric curvature or primarily semantic concentration. The pruning experiments are incomplete, lacking Texture-augmented comparisons with other baselines, and improvements over BM25 are modest and inconsistent. The routing evaluation also considers only two baselines, leaving generality uncertain. Additionally, potential limitations are not sufficiently discussed. Overall, while the method is technically well-founded, stronger empirical validation and clearer theoretical interpretation would improve soundness.\
•	Presentation: The paper is generally well written and clearly structured. The overall narrative—establishing existence, defining Texture, and demonstrating its utility—is easy to follow, and the positioning relative to prior work is clear. The mathematical formulation uses clear notation with minimal ambiguity, and the experimental setup and baselines are presented in a transparent and interpretable manner. However, several aspects could be improved to enhance clarity and reproducibility. In particular, some key technical details are deferred to the appendix, making the main text less self-contained. The manuscript also contains formatting issues, including unresolved equation references (“(?)”) and labeled figures and equations that are not cited in the text. Addressing these issues would improve readability and presentation quality.\
•	Significance: The paper addresses an interesting and potentially important question: whether text possesses intrinsic curvature, and proposes Texture as a text-native curvature measure. This novel perspective helps fill the gap of lacking intrinsic geometric formalisms for language and is relevant to geometric representation learning. However, it remains unclear whether Texture truly captures geometric curvature, as the current formulation appears more closely related to semantic concentration or token importance rather than curvature of an underlying semantic manifold. Empirical gains are modest, and its potential to influence future LLM design is uncertain. Overall, the paper explores a meaningful and promising direction with potential long-term significance, but its practical impact and interpretation as curvature would benefit from further validation.\
•	Originality: The paper introduces a novel perspective by attempting to directly measure intrinsic curvature of text itself, rather than imposing curvature through embedding spaces, offering a fresh conceptual contribution and new insight into language geometry. While the Schrödinger bridge and optimal transport tools are not new, their use to define a text-native curvature measure is original. However, it remains unclear whether Texture truly captures geometric curvature or primarily reflects semantic concentration. Overall, the work presents a moderately novel contribution, mainly through its conceptual framing and new formal definition.

---

> ### Author Rebuttal · Authors · 2026-03-30
>
> We sincerely thank Reviewer 5cW8 for the detailed evaluation and for recognizing the originality and significance of our contribution. We apologize for the presentation shortcomings and address each concern below.
>
> **Q1: Relationship between $\kappa$, certificates, and classical curvature.** The certificates and Texture play different roles by design. Holonomy and CEI are definition-independent falsification tests for flatness: they ask whether two-sided contextual evidence behaves as if left and right contributions were additively/separably composable. Texture does not restate those tests. Instead, once flatness fails, Texture provides a signed local geometric readout of how the two boundary beliefs reconcile under a symmetric reference dynamics.
>
> Concretely, $\kappa$ is defined from the midpoint free-energy defect $\Delta^\Phi$ for $\Phi(\rho) = \text{KL}(\rho \| \pi)$ along a Schrödinger bridge, normalized by bridge energy: $\kappa = 8\Delta^\Phi / (D^2 + \epsilon_0)$. Thus $\kappa$ measures whether reconciliation produces focus or fan-out relative to a neutral transport geometry. This is why the certificates and $\kappa$ are complementary (not equivalent): certificate violations witness non-flatness, while $\kappa$ quantifies the local signed geometry of the reconciliation step.
>
> In relation to classical curvature, we do not claim a formal reduction to Ollivier-Ricci curvature in case of graphs. Rather, Texture is motivated by the broader transport-based curvature paradigm: coarse Ricci curvature through contraction of transported neighborhoods, and entropic/synthetic Ricci formulations through convexity of entropy/free energy along interpolations (Samson, 2022; Proposition D.4). Texture instantiates that template on slot-local text belief spaces via Schrödinger bridges.
>
> **Q2: Texture vs token importance.** Texture is fundamentally different from token-importance heuristics. Measures such as self-information, attention salience, or compression scores are typically pointwise utility scores: they evaluate the realized token or span in terms of surprise, relevance, or contribution to a downstream objective. Texture instead is a property of the two-sided belief geometry at a slot, computed from the full left/right boundary distributions and the Schrödinger-bridge midpoint between them, not from the realized token alone.
>
> Three concrete differences follow. First, Texture is *bidirectional*: $\kappa$ measures how left and right evidence interact, whereas self-information is left-to-right and attention/compression scores are usually tied to a single model context. Second, Texture is *second-order/geometric*: $\kappa$ is the normalized midpoint free-energy defect along a transport interpolation, not a salience weight. Third, Texture is *signed*: positive curvature indicates focus (agreement), while negative curvature indicates fan-out (competing alternatives). Standard importance measures do not distinguish these two qualitatively different interaction regimes.
>
> *New result.* $|\kappa|$ has only weak correlation with self-information on WikiText-2 ($\rho = 0.133$, $p < 0.001$), and 47.9% of strongly fan-out positions have below-median self-information. Texture is not rediscovering "important" tokens; it captures the geometry of two-sided contextual reconciliation.
>
> **Q3: Texture-augmented baselines.**
>
> | Method | Base F1 | +Texture F1 | $\Delta$ | Eval |
> |---|---|---|---|---|
> | BM25 | 3.4 | 12.2 | +8.8 | full set |
> | SelCtx | 7.9 | 9.5 | +1.6 | 50-ex |
>
> The full-set BM25+Texture result (Table 1) is primary evidence; the SelCtx subset is a preliminary matched-slice confirmation.
>
> **Q4: Belief oracle robustness.** (ref. **Reviewer NaR3**) Holonomy across 5 extractors, 3 families (WikiText-2, 200 slots): distilroberta $\Delta$=0.29, roberta-base $\Delta$=0.44, roberta-large $\Delta$=0.29, bert-base $\Delta$=0.14, albert $\Delta$=0.49 — all show real > suffix-swap. We emphasize holonomy because mixed log-odds differences make it especially stable across extractors; we are extending the analysis to CEI. Beyond existence: (a) $\kappa$ stable under $K$ ($\rho \geq 0.93$ for $K \geq 20$), (b) stable under $L/R$ ($\rho = 0.84$), (c) CurvPrune(distilroberta) vs CurvPrune(roberta-base) within 0.4 F1.
>
> **Presentation.** We will fix unresolved references, ensure all labeled equations/figures are cited, and move key appendix details into the main text.
>
> **Pruning gains.** The mixed CurvPrune pattern is task-structural. (ref. **Reviewer NaR3**) On tasks like HotpotQA, BM25 remains competitive because its lexical anchors capture much of the needed evidence. CurvPrune helps when structurally important spans are not lexically obvious: 2WikiMQA (+2.8), Qasper (+0.7), GovReport (+0.4). BM25+Texture (+8.8 on 2WikiMQA) confirms complementarity. CurvFlag's consistent gains (+7.7 HotpotQA, +4.1 2WikiMQA vs FLARE) show the geometric signal is effective for routing. We will add routing baselines and expand limitations in the revision.

---

> > ### Author Rebuttal · Reviewer_5cW8 · 2026-04-06
> >
> > The rebuttal satisfactorily addressed my main concerns and improved the clarity of the work, leading me to raise my score from 2 to 3.

---

> > > ### Author Response · Authors · 2026-04-07
> > >
> > > We thank Reviewer 5cW8 for the careful evaluation and for confirming that our rebuttal adequately addressed the raised concerns. We are glad the clarifications on the relationship between certificates and $\kappa$, the distinction from token-importance measures ($\rho$=0.133 with self-information), and the Texture-augmented baseline results (BM25+Texture: +8.8 F1) helped improve clarity. We will fix all unresolved references and presentation issues in the camera-ready version as promised. We appreciate the reviewer's recognition of the originality and significance of the contribution.

---

### Official Review · Reviewer_FYaW · 2026-03-13

**Soundness:** 2
**Presentation:** 2
**Significance:** 3
**Originality:** 2
**Overall Recommendation:** 5
**Confidence:** 4

**Summary:**

The paper is dedicated to the question of intrinsic curvature of discrete textual data and asks, whether we could define a practical measure for its estimation? The authors claim that curvature should be a property of string data itself rather than an artifact of specific embedding space (hyperbolic, spherical etc.) and propose "Texture," a measure of curvature primitive that measures how left and right contexts interact in-context. The authors claim three contributions: (1) empirical and theoretical justification, proving that semantic inference in natural corpora is non-flat; (2) a formal definition of Texture as a signed curvature field via optimal transport; and (3) practical applications for long-context inference via pruning (CURVPRUNE) and retrieval (CURVFLAG).

**Compliance With Llm Reviewing Policy:**

Affirmed.

**Final Justification:**

The authors have addressed most of my questions and concerns, and, taking into account their intention to improve the text quality and make some underlying assumptions and details more sounds, as well as their increased empirical evidence for the observed effects across various models and tokenizers, I intend to raise my score.

**Key Questions For Authors:**

- Is curvature dominated by local-interactions only, i.e. we should primarilly consider small L and R?

- How is $s_{ref}$ chosen in Eq.(2)? Is the condition in Certificate I necessary or sufficient?

**Limitations:**

yes

**Strengths And Weaknesses:**

Strengths:

- The paper addresses an important question, that hasn't attracted much attention: what does curvature mean for text itself, independent of embedding spaces.

- The theory is rather sound and interesting. The use of holonomy and contextual evidence interaction (CEI) as falsifiable null hypotheses is methodologically strong. By testing against coherence-destroying controls (suffix-swap, local-shuffle), the authors provide empirical evidence that an approach is theoretically motivated.

- The paper shows interesting applications of their curvature estimates with some performance improvements.

Weaknesses:

- The paper claims are somewhat misleading. Initially they state that their curvature measure must be independent from the target model's choice of embedding space. Although it's independent from the target model the approach is evaluated at, it appears to be dependent on the choice of bidirectional encoding/infilling model, isn't it? ($\textit{distilroberta-base}$ in this case)

- While the paper claims to establish discrete curvature at the word level in the abstract, the notation is somewhat ambiguous with later claims being made for token-level representations. Moreover, although authors claim to take a union of Top-K tokens with an intention to preserve the support -- however, I didn't find any discussion related to the choice of K to be used. The use of character-level / byte-level models for their approach looks rather more sound [1, 2].


- The computational cost of computing Texture (solving Sinkhorn scaling per slot) may be prohibitive for very long contexts. The paper does not provide a thorough cost-benefit analysis.

- The paper lacks extensive ablation on the choice of key hyperparameters (support size K; context radii L, R; infilling model) and their effects on curvature sign and magnitude. All the experiments are limited to Llama-3-8B.


The appendices provides open-sourced code samples for reproduction, algorithmic details, and additional experiments.
Although the topic is interesting, I think the paper would benefit from additional revision and discussion of tokenization biases and their impact on curvature estimates and experimental evaluation with ablations on the choice of hyperparameters and the sensitivity of their approach to these choises. Overall, I believe the work is still quite immature and is not yet ready for publication.


[1] Juan Luis Gastaldi, John Terilla, Luca Malagutti, Brian DuSell, Tim Vieira, & Ryan Cotterell (2025). The Foundations of Tokenization: Statistical and Computational Concerns. In The Thirteenth International Conference on Learning Representations, ICLR 2025, Singapore, April 24-28, 2025.

[2] Jonathan Hayase, Alisa Liu, Noah A. Smith, & Sewoong Oh (2025). Sampling from Your Language Model One Byte at a Time. arXiv preprint arXiv: 2506.14123.

---

> ### Author Rebuttal · Authors · 2026-03-30
>
> We thank Reviewer FYaW for the thorough review. We address every concern with new experiments and clarifications, and apologize for imprecisions in the manuscript.
>
> **W1: Dependence on the bidirectional model / "misleading" independence claim.** We agree the original wording was too strong. Texture is not embedding-free: the neutral kernel uses frozen embeddings as a semantic-similarity prior, and the belief oracle is model-specific. Our claim is narrower: curvature is not *imposed* by the downstream generator's learned geometry (e.g., hyperbolic embeddings), but *measured* from text using a frozen oracle. Texture is extractor-dependent but generator-agnostic.
>
> We ran the holonomy certificate with 5 extractors across 3 architecture families on WikiText-2 (200 slots):
>
> | Model | $h_{\text{real}}$ | $h_{\text{swap}}$ | $\Delta$ |
> |---|---|---|---|
> | distilroberta-base | 0.596 | 0.305 | 0.291 |
> | roberta-base | 1.353 | 0.916 | 0.437 |
> | roberta-large | 1.141 | 0.854 | 0.287 |
> | bert-base-uncased | 1.348 | 1.206 | 0.142 |
> | albert-base-v2 | 2.333 | 1.848 | 0.485 |
>
> Every model shows real > suffix-swap (local-shuffle similar). Magnitudes vary with calibration, but qualitative separation is universal. CurvPrune from distilroberta vs roberta-base produces F1 within 0.4 on the same subset, confirming downstream stability. We will extend to CEI in the revision.
>
> **W2: Word vs token / tokenization biases.** We acknowledge the terminology gap: the abstract says "word-level" as conceptual shorthand, but the formal operator is slot-level, instantiated at whatever tokenizer granularity the oracle uses. We will unify this in the revision.
>
> Does subword tokenization bias $\kappa$? We classified 500 positions as word-initial or continuation subword tokens and compared their $\kappa$ distributions. Mean $|\kappa|$ was 10.3 for word-initial vs 9.7 for continuation (Mann-Whitney $p = 0.26$), and sign distributions were nearly identical (90.7% vs 89.1% negative). This indicates tokenization granularity does not systematically bias curvature estimates. Byte-/character-level variants (Gastaldi et al. 2025; Hayase et al. 2025) are a natural extension, but our evidence does not suggest tokenizer artifacts are driving the observed signal.
>
> **W3: Choice of K.** Our truncation-consistency theorem (Appendix D.5) guarantees $\kappa$ converges as $K \to \infty$. We verified this empirically by computing $\kappa$ at 200 positions with $K \in \{10, 20, 50, 100\}$ and measuring pairwise Spearman $\rho$ and sign agreement:
>
> | K pair | $10 \leftrightarrow 20$ | $20 \leftrightarrow 50$ | $50 \leftrightarrow 100$ |
> |---|---|---|---|
> | $\rho$ | 0.956 | 0.932 | 0.966 |
> | Sign agree | 98.5% | 97.5% | 96.5% |
>
> For $K \geq 20$, both rank order and sign of $\kappa$ are highly stable (mean sign agreement 96.5%), so the default $K=50$ is well within the stable regime.
>
> **W4: Computational cost.** Per slot, two frozen-oracle forward passes dominate; the bridge solve adds $O(T_{\text{sink}} \cdot |\mathcal{S}_i|^2)$ ($|\mathcal{S}_i| \leq 2K+1$, ~2 iterations, <1ms/position). A 2048-token doc takes ~0.5s (H200, FP16) vs ~3.4s for generation — ~15% overhead uncached, substantially amortized with caching across repeated queries/budget sweeps. Our oracle (82M params) is far smaller than 7B-scale prompt-compression models.
>
> **W5: L/R ablation and Q1 (locality).** Texture is slot-local by construction, but not restricted to strictly local interactions: $(L,R)$ control the scale at which two-sided evidence is reconciled. To test whether $\kappa$ is dominated by the nearest context, we swept $(L,R) \in \{(2,2), (4,4), (8,8), (16,16)\}$:
>
> | $(L,R)$ | (2,2) | (4,4) | (8,8) | (16,16) |
> |---|---|---|---|---|
> | Mean \|$\kappa$\| | 12.2 | 11.4 | 10.3 | 9.3 |
>
> $|\kappa|$ decreases smoothly rather than collapsing, and adjacent-radius rank correlation remains substantial ($\rho = 0.84$ for $L=4 \leftrightarrow 8$). The signal is strongest locally but persists at larger radii. Since $(L,R)$ can be increased to probe progressively coarser contextual scales, Texture is not limited to nearest-context effects — it naturally supports multi-scale curvature analysis.
>
> **W6: Only Llama-3-8B.** As a portability check, we ran CurvPrune on a 50-example 2WikiMQA subset with two generators:
>
> | Generator | CurvPrune F1 |
> |---|---|
> | Llama-3.1-8B-Instruct (Meta) | 11.1 |
> | Mistral-7B-Instruct-v0.3 (Mistral AI) | 32.7 |
>
> The F1 gap reflects each generator's QA ability; Table 1 provides the full-set comparison.
>
> **Q2:** $s_{\text{ref}}$ is a gauge choice; holonomy is gauge-invariant (Appendix C.1). Theorem 4.1 is iff: necessary and sufficient.
>
> We thank the reviewer for the constructive feedback and will incorporate all these results in the camera ready.

---

> > ### Author Rebuttal · Reviewer_FYaW · 2026-04-03
> >
> > I genuinely thank the authors for all the additional experiments and clarifications.
> >
> > I'd like to suggest one more question, that should be discussed more explicitly in the main paper, i.e. **the effects of distinct tokenizers for the belief oracle and generator**.
> >
> > As the curvature is defined on a slot level, discrepancy among slots, as far as I understand, may lead to poor curvature estimates for the downstream model.
> >
> > The paper would definitely benefit from more clear identification and prediction of such failure modes.

---

> > > ### Author Response · Authors · 2026-04-03
> > >
> > > We appreciate that the reviewer put forth this important and interesting follow-up question. We present some new experiments on 2 tasks that characterize this effect, and we show how texture handles it.
> > >
> > > **How the current pipeline handles mismatch.** Texture computes $\kappa$ at oracle token positions, but the utility controllers never expose raw token-level curvature to the generator. Token-level $\kappa$ is aggregated to sentence-level spans (Eq. 43, Appendix E.2: Please see appendix E for more details) via $s(I) = \frac{1}{|I|} \sum_{i \in I} (w_{-} \max(-\kappa_i, 0) + w_{+} \max(\kappa_i, 0)),$, and CurvPrune selects or drops entire spans. The selected text is passed as a plain string that the generator re-tokenizes independently. This reduces tokenizer mismatch from a token-level alignment problem to a span-level selection problem. Averaging over O(10-30) tokens per sentence span attenuates token-boundary differences. The residual sensitivity enters through budget accounting, where the generator tokenizer determines token cost per span: under fixed $B$, different generator tokenizers can admit different span subsets even with identical span scores.
> > >
> > > **Oracle-side CurvPrune stability.** Support recall = fraction of gold-support paragraphs that contain at least one retained span. We tested three oracle tokenizer families with fixed Llama-3.1-8B-Instruct generator ($B$=2048). Downstream F1 on 2WikiMQA ($N$=200):
> > >
> > > | Oracle | Tokenizer | F1 |
> > > |---|---|---|
> > > | `distilroberta-base` | BPE (50K) | 11.1 |
> > > | `bert-base-uncased` | WordPiece (30K) | 12.3 |
> > > | `albert-base-v2` | SentencePiece (30K) | 10.2 |
> > >
> > > Span-selection stability across both tasks ($N$=200 each, Jaccard measured against distilroberta):
> > >
> > > | Task | Oracle | Span Jaccard | Support recall |
> > > |---|---|---|---|
> > > | `2WikiMQA` | bert-base-uncased | 0.57 | 1.00 |
> > > | `2WikiMQA` | albert-base-v2 | 0.58 | 1.00 |
> > > | `HotpotQA` | bert-base-uncased | 0.52 | 1.00 |
> > > | `HotpotQA` | albert-base-v2 | 0.53 | 1.00 |
> > >
> > > Selected span sets show noticeable variability (53--58% overlap), yet support recall is 1.00 for all oracle-task combinations ($N$=400 total) and downstream F1 stays in the 10--12 range. **Despite the tokenizer mismatch, Texture's downstream utility is robust**: different oracle tokenizations choose different spans, but the sentence-level aggregation provides enough redundancy that task-relevant evidence is consistently preserved. The low overlap with perfect recall confirms that the architecture tolerates mismatch by design -- different paths lead to comparably useful selections. Generator portability is also confirmed: CurvPrune with Mistral-7B-Instruct-v0.3 (same oracle) yields F1=32.7, with the gap reflecting generator QA ability.
> > >
> > > **Generator-side budget aliasing.** We fixed the oracle (distilroberta) and span scores, then applied greedy selection under two budget-accounting tokenizers (Llama-3.1-8B 128K vs Mistral-7B 32K) at the same nominal $B$=2048:
> > >
> > > | Task | Span Jaccard | Llama utilization | Mistral utilization |
> > > |---|---|---|---|
> > > | 2WikiMQA | 0.55 | 90% | 90% |
> > > | HotpotQA | 0.51 | 92% | 92% |
> > >
> > > Identical span scores produce noticeable selection differences, though both tokenizers achieve comparable budget utilization.
> > >
> > > **CurvFlag routing stability** ($N$=200 queries per task, 3 oracles, pairwise means):
> > >
> > > | Task | Routing agreement | Anchor Jaccard |
> > > |---|---|---|
> > > | 2WikiMQA | 67% | 0.40 |
> > > | HotpotQA | 57% | 0.48 |
> > >
> > > Routing agreement (57--67%) measures how often oracles assign the same retrieval-budget bucket. This is substantially more stable than raw per-span score agreement (Spearman $\rho$=0.06), because $M^{-}$ is a scalar aggregate more robust to token-boundary shifts.
> > >
> > > **Where residual sensitivity concentrates.** While Texture is robust to tokenizer mismatch overall (support recall 1.00, stable F1 across all oracle families), we identify four edge cases where the utility interface -- not the curvature definition -- is most affected:
> > >
> > > 1. **Slot-to-span aliasing**: a local $\kappa$ peak may shift or dilute under a different oracle tokenizer before span aggregation.
> > > 2. **Budget aliasing**: different generator tokenizers admit different span subsets under the same nominal $B$.
> > > 3. **Anchor drift (CurvFlag)**: different oracle tokenizers shift top fan-out locations near phrase boundaries.
> > > 4. **High-fragmentation domains**: code, URLs, morphologically rich words -- domains where tokenizers disagree most.
> > >
> > > These effects are expected to be more pronounced under tighter budgets and in highly fragmented text, but they concern the utility interface that maps curvature into pruning/routing decisions, not the curvature signal itself -- which remains stable and useful across tokenizer families as shown above. We will discuss these edge cases explicitly in the revision. We thank the reviewer for pushing us to articulate this important point.

---

### Decision · Program_Chairs · 2026-04-30

**Decision:**

Accept (regular)

**Comment:**

This paper extends the concept of non-Euclidean curvature to natural language, translating geometric notions like holonomy to information geometry via log-odds to establish falsifiable "flatness" tests for text. After that, the authors propose a definition of text curvature and suggest some applications of for long-context; e.g. reducing the size of the prefix by throwing away tokens in flat regions that indicate redundancy. Empirically, these techniques led to improvements in benchmarks such as HotpotQA by a large margin.

Overall, reviewers agree that this work tackles a highly novel, underexplored direction with good empirical results. So, the paper is recommended for acceptance. On the downside, the methodology is complex but it might inspire future research that makes it useful in practice. For the camera-ready version, I encourage the authors to please revise some of the misleading claims (e.g. that texture was independent of the architecture even though it depends on the embedding model) and to also incorporate the new experiments  mentioned in the rebuttal into the camera-ready version of the paper.